# Understanding Gradient Regularization in Deep Learning: Efficient Finite-Difference Computation and Implicit Bias

## Abstract

Gradient regularization (GR) is a method that penalizes the gradient norm of the training loss during training. Although some studies have reported that GR improves generalization performance in deep learning, little attention has been paid to it from the algorithmic perspective, that is, the algorithms of GR that efficiently improve performance. In this study, we first reveal that a specific finite-difference computation, composed of both gradient ascent and descent steps, reduces the computational cost for GR. In addition, this computation empirically achieves better generalization performance. Next, we theoretically analyze a solvable model, a diagonal linear network, and clarify that GR has a desirable implicit bias in a certain problem. In particular, learning with the finite-difference GR chooses better minima as the ascent step size becomes larger. Finally, we demonstrate that finite-difference GR is closely related to some other algorithms based on iterative ascent and descent steps for exploring flat minima: sharpness-aware minimization and the flooding method. We reveal that flooding performs finite-difference GR in an implicit way. Thus, this work broadens our understanding of GR in both practice and theory.

## 1 Introduction

Explicit or implicit regularization is a key component for achieving better performance in deep learning. For instance, adding some regularization on the local sharpness of the loss surface is one common approach to enable the trained model to achieve better performance (Hochreiter & Schmidhuber, 1997; Foret et al., 2021; Jastrzebski et al., 2021). In the related literature, some recent studies have empirically reported that gradient regularization (GR), i.e., adding penalty of the gradient norm to the original loss, makes the training dynamics reach flat minima and leads to better generalization performance (Barrett & Dherin, 2021; Smith et al., 2021; Zhao et al., 2022). Using only the information of the first-order gradient seems a simple and computationally friendly idea. Because the first-order gradient is used to optimize the original loss, using its norm is seemingly easier to use than other sharpness penalties based on second-order information such as the Hessian and Fisher information (Hochreiter & Schmidhuber, 1997; Jastrzebski et al., 2021).

Despite its simplicity, our understanding of GR has been limited so far in the following ways. First, we need to consider the fact that GR must compute *the gradient of the gradient* with respect to the parameter. This type of computation has been investigated in a slightly different context: input-Jacobian regularization, that is, penalizing the gradient with respect to the input dimension to increase robustness against input noise (Drucker & Le Cun, 1992; Hoffman et al., 2019). Some studies proposed the use of double backpropagation (DB) as an efficient algorithm for computing the gradient of the gradient for input-Jacobian regularization, whereas others proposed the use of finite-difference computation (Peebles et al., 2020; Finlay & Oberman, 2021). Second, theoretical understanding of GR has been limited. Although empirical studies have confirmed that the GR causes the gradient dynamics to eventually converge to better minima with higher performance, the previous work provides no concrete theoretical evaluation for this result. Third, it remains unclear whether the GR has any potential connection to other regularization methods. Because the finite difference is composed of both gradient ascent and descent steps by definition, we are reminded of some learning algorithms for exploring flat minima such as sharpness-aware minimization (SAM) (Foret et al.,

2021) and the flooding method (Ishida et al., 2020), which are also composed of ascent and descent steps. Clarifying these points would help to deepen our understanding on efficient regularization methods for deep learning.

In this work, we reveal that GR works efficiently with a finite-difference computation. This approach has a lower computational cost, and surprisingly achieves better generalization performance than the other computation methods. We present three main contributions to deepen our understanding of GR:

- We demonstrate some advantages to using the finite-difference computation. We give a brief estimation of the computational costs of finite difference and DB in a deep neural network and show that the finite difference is more efficient than DB (Section 3.2). We find that a so-called forward finite difference leads to better generalization than a backward one and DB (Section 3.3). Learning with forward finite-difference GR requires two gradients of the loss function, gradient ascent and descent. A relatively large ascent step improves the generalization.

- We give a theoretical analysis of the performance improvement obtained by GR. we analyze the selection of global minima in a diagonal linear network (DLN), which is a theoretically solvable model. We prove that GR has an implicit bias for selecting desirable solutions in the so-called rich regime (Woodworth et al., 2020) which would potentially lead to better generalization (Section 4.2). This implicit bias is strengthened when we use forward finite-difference GR with an increasing ascent step size. In contrast, it is weaken for a backward finite difference, i.e., a negative ascent step.

- Finite-difference GR is also closely related to other learning methods composed of both gradient ascent and descent, that is, SAM and the flooding method. In particular, we reveal that the flooding method performs finite-difference GR in an implicit way (Section 5.2).

Thus, this work gives a comprehensive perspective on GR for both practical and theoretical understanding.

## 2 RELATED WORK

Barrett & Dherin (2021) and Smith et al. (2021) investigated explicit and implicit GR in deep learning. They found that the discrete-time update of the usual gradient descent implicitly regularizes the gradient norm when its dynamics are mapped to the continual-time counterpart. This is referred to as implicit GR. They also investigated explicit GR, i.e., adding a GR term explicitly to the original loss, and reported that it improved generalization performance even further. Jia & Su (2020) also empirically confirmed that the explicit GR gave the improvement of generalization. Barrett & Dherin (2021) characterized GR as the slope of the loss surface and showed that a low GR (gentle slope) prefers flat regions of the surface. Recently, Zhao et al. (2022) independently proposed a similar but different gradient norm regularization, that is, explicitly adding a non-squared L2 norm of the gradient to the original loss. They used a forward finite-difference computation, but its superiority to other computation methods remains unconfirmed.

The implementation of GR has not been discussed in much detail in the literature. In general, to compute the gradient of the gradient, there are two well-known computational methods: DB and finite difference. Some previous studies applied DB to the regularization of an information matrix (Jastrzebski et al., 2021) and input-Jacobian regularization, i.e., adding the L2 norm of the derivative with respect to the input dimension (Drucker & Le Cun, 1992; Hoffman et al., 2019). Others have used the finite-difference computation for Hessian regularization (Peebles et al., 2020) and input-Jacobian regularization (Finlay & Oberman, 2021). Here, we apply the finite-difference computation to GR and present some evidence that the finite-difference computation outperforms DB computation with respect to computational costs and generalization performance.

In Section 4, we give a theoretical analysis of learning with GR in *diagonal linear networks* (DLNs) (Woodworth et al., 2020). The characteristic property of this solvable model is that we can evaluate the implicit bias of learning algorithms (Nacson et al., 2022; Pesme et al., 2021). Our analysis includes the analysis of SAM in DLN as a special case (Andriushchenko & Flammarion, 2022). In contrast to previous work, we evaluate another lower-order term, and this enables us to show that forward finite-difference GR selects global minima in the so-called rich regime.

## 3 GRADIENT REGULARIZATION

We consider GR (Barrett & Dherin, 2021; Smith et al., 2021), wherein the squared L2 norm of the gradient is explicitly added to the original loss $\mathcal{L}(\theta)$ as follows:

$$\tilde{\mathcal{L}}(\theta) = \mathcal{L}(\theta) + \frac{\gamma}{2}R(\theta), \quad R(\theta) = \|\nabla\mathcal{L}(\theta)\|^2, \tag{1}$$

where $\|\cdot\|$ denotes the Euclidean norm and $\gamma > 0$ is a constant regularization coefficient. We abbreviate the derivative with respect to the parameters $\nabla_\theta$ by $\nabla$. Its gradient descent is given by

$$\theta_{t+1} = \theta_t - \eta\nabla\tilde{\mathcal{L}}(\theta_t) \tag{2}$$

for time step $t = 0, 1, ...$ and learning rate $\eta > 0$. While previous studies have reported that explicitly adding a GR term empirically improves generalization performance, its algorithms and implementations have not been discussed in much detail.

### 3.1 ALGORITHMS

To optimize the loss function with GR (1) using a gradient method, we need to compute the gradient of the gradient, i.e., $\nabla R(\theta)$. As is well studied in input-Jacobian regularization (Drucker & Le Cun, 1992; Hoffman et al., 2019; Finlay & Oberman, 2021), there are two main approaches to computing the gradient of the gradient.

**Finite difference:** The finite-difference method approximates a derivative by a finite step. In the case of GR, we have $\nabla R(\theta_t)/2 = (\nabla\mathcal{L}(\theta') - \nabla\mathcal{L}(\theta_t))/\varepsilon + \mathcal{O}(\varepsilon)$ with $\theta' = \theta_t + \varepsilon\nabla\mathcal{L}(\theta_t)$ for a constant $\varepsilon > 0$. The final term is expressed in Landau notation and is neglected in the computation. We update the GR term by

$$\Delta R_F(\varepsilon) = \frac{\nabla\mathcal{L}(\theta_t + \varepsilon\nabla\mathcal{L}(\theta_t)) - \nabla\mathcal{L}(\theta_t)}{\varepsilon} \quad \text{(F-GR)}. \tag{3}$$

We refer to this gradient as *Forward finite-difference GR (F-GR)*. Because the gradient $\nabla\mathcal{L}(\theta_t)$ is computed for the original loss, the finite difference (3) requires only one additional gradient computation $\nabla\mathcal{L}(\theta')$. The order of the computation time is only double that of the usual gradient descent. The finite-difference method also has a backward computation:

$$\Delta R_B(\varepsilon) = \frac{\nabla\mathcal{L}(\theta_t) - \nabla\mathcal{L}(\theta_t - \varepsilon\nabla\mathcal{L}(\theta_t))}{\varepsilon} \quad \text{(B-GR)}. \tag{4}$$

If we allow a negative step size, $\Delta R_B$ corresponds to $\Delta R_F$ through $\Delta R_B(\varepsilon) = \Delta R_F(-\varepsilon)$. For a sufficiently small $\varepsilon$, both finite-difference GRs yield the same original gradient $\nabla R(\theta)$ if we can neglect any numerical instability caused by the limit. The finite-difference method has been used in the literature for the optimization of neural networks, especially for Hessian-based techniques (Bishop, 2006; Peebles et al., 2020). When we need a more precise $\nabla R$, we can use a higher-order approximation, e.g., the centered finite difference, but this requires additional gradient computations, and hence we focus on the first-order finite difference.

**Double Backpropagation:** The other approach is to apply the automatic differentiation directly to the GR term, i.e., $\nabla R$. For example, its PyTorch implementation is quite straightforward, as shown in Section C.1 of the Appendices. This approach is referred to as DB, which was originally developed for input-Jacobian regularization (Drucker & Le Cun, 1992). We explain more details on the DB computation and its computational graph in Section 3.2. DB, in effect, corresponds to computing the following Hessian-vector product:

$$\Delta R_{DB} = H(\theta_t)\nabla\mathcal{L}(\theta_t), \tag{5}$$

where $H(\theta) = \nabla\nabla\mathcal{L}(\theta)$. The following equation may give us an intuition about the difference between the finite difference and DB alternatives. From the mean value theorem, F-GR is equivalent to

$$\Delta R_F(\varepsilon) = \frac{1}{\varepsilon}\int_0^\varepsilon ds H(\theta_t + s\nabla\mathcal{L}(\theta_t))\nabla\mathcal{L}(\theta_t). \tag{6}$$

We can interpret the finite difference as taking an average of the curvature (Hessian) along the line of gradient update. For $\varepsilon \to 0$, this reduces to $\Delta R_{DB}$.

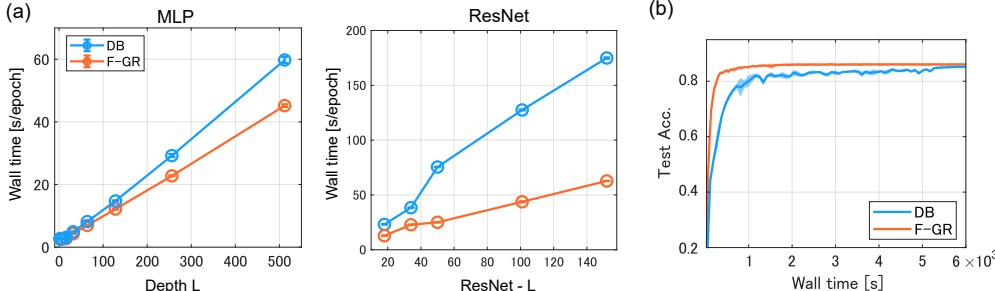

Figure 1: Finite-difference computation is more efficient than DB computation in wall time. (a) Wall time required for learning with GR in one epoch. For the ResNet, we used ResNet-$\{18, 34, 50, 101, 152\}$. (b)Training dynamics in ResNet-18 on CIFAR-10. Learning with F-GR is much faster in wall time.

Note that the difference among these algorithms appears in non-linear models. For a naive linear model $X\theta$, the squared error loss has a constant Hessian $XX^\top$. Therefore, all of $\Delta R$ have the same update. We analyze a simple network model with non-linearity on the parameters in Section 4 and reveal the difference of implicit biases.

### 3.2 COMPUTATIONAL COST

We clarify the computational efficiencies of each algorithm of GR in deep networks. First, we give a rough estimation of the computational cost by counting the number of matrix multiplication required to compute $\nabla\tilde{\mathcal{L}}$. Consider an $L$-layer fully connected neural network with a linear output layer: $A_l = \phi(U_l)$, $U_l = W_l A_{l-1}$ for $l = 1, ..., L$. Note that $A_l$ denotes a batch of activation and $W_l A_{l-1}$ requires a matrix multiplication. We denote the element-wise activation function as $\phi(\cdot)$ and weight matrix as $W_l$. For simplicity, we neglect the bias terms. The number of matrix multiplications required to compute $\nabla\tilde{\mathcal{L}}$ is given by

$$N_{mul} \sim 6L \text{ (for F-GR)}, \quad 9L \text{ (for DB)}, \tag{7}$$

where $\sim$ hides an uninteresting constant shift independent of the depth. One can evaluate $N_{mul}$ straightforwardly from the computational graph (Figure 2), originally developed for the DB computation of input-Jacobian regularization (Drucker & Le Cun, 1992). In brief, the original gradient $\nabla\mathcal{L}$, that is, the backpropagation on the forward pass $\{A_0 \to A_1 \to \cdots \to A_L\}$, requires $3L$ matrix multiplications: $L$ for the forward pass, $L$ for backward pass $B_l = \phi'(U_l)\circ(W_{l+1}^\top B_{l+1})$, and $L$ for gradient $G_l := \partial\mathcal{L}/\partial W_l = B_l A_{l-1}^\top$. Because F-GR is composed of both gradient ascent and descent steps, we eventually need $6L$. In contrast, for learning using the DB of GR, we need $3L$ for $\nabla\mathcal{L}$ and additional $6L$ for the GR term. The GR term requires a forward pass of composed of $A_l$, $B_l$, and $G_l$ obtained in the gradient computation of $\nabla\mathcal{L}$. Note that the upper part $\{A_0 \to A_1 \to \cdots \to B_L \to \cdots \to B_1\}$ is well known as the DB of input-Jacobian regularization. As pointed out in Drucker & Le Cun (1992), the computation of $\nabla B_1$ is equivalent to treating the upper part of the graph as the forward pass and applying backpropagation. It requires $2L$ multiplications. In our GR case, we have additional $L$ multiplications due to $G_l$. Because the backward pass doubles the number of required multiplications, we eventually need $2 \times (2L + L) = 6L$ multiplication. Further details are given in Section C.1.

The results of numerical experiments shown in Figure 1 confirm the superiority of finite-difference GR in typical experimental settings. We trained deep neural networks using an NVIDIA A100 GPU for this experiment. All experiments were implemented by PyTorch. We summarize the pseudo code and implementation of GR and present the detailed settings of all experiments in Section C. Figure 1(a) shows the wall time required for one epoch of training with stochastic gradient descent (SGD) and the objective function (1). We trained various multi-layer perceptrons (MLPs) and residual neural networks (ResNets) with different depths. The wall time increased almost linearly as the depth increased. The slope of the line is different for F-GR and DB, and F-GR was faster. This observation is consistent with the number of multiplications (7). In particular, in ResNet, one of the most typical deep neural networks, learning with finite-difference GR was more than twice as fast as learning with DB. Figure 1(b) confirms that F-GR has fast convergence in ResNet-18 on CIFAR-10. In Figure S.1

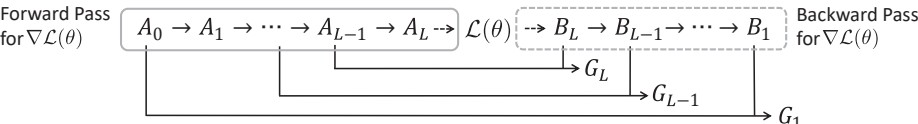

Figure 2: Computational graph of DB. Each node with an incoming solid arrow requires one matrix multiplication for the forward pass.

, we also show the convergence measured by the training loss and time steps. All of them showed better convergence for the finite difference.

Note that the finite difference is also better to use from the perspective of memory efficiency. This is because DB requires all of the $\{A_l, B_l, G_l\}$ to be retained for the forward pass, which occupies more memory. It is also noteworthy that in general, it is difficult for theory to completely predict the realistic computational time required because it could heavily depend on the hardware and the implementation framework and does not necessarily correlate well with the number of floating-point operations (FLOPs) (Dehghani et al., 2021). Our result suggests that at least the number of matrix multiplication explains well the superiority of the finite-difference approach in typical settings.

### 3.3 Generalization performance

Here, we show that the superiority of finite-difference computation over DB also appears in the eventual performance of trained models. Figures 3 and S.2 show the test accuracy of a 4-layer MLP and ResNet-18 trained by using SGD with GR on CIFAR-10 We trained the models in an exhaustive manner with various values for $\gamma$ and $\varepsilon$ for each algorithm of the GR. For learning with F-GR, the model achieved the highest accuracy on relatively large ascent steps ($\varepsilon \sim 0.1$). In contrast, learning with B-GR showed a rapid decrease of the performance as the step size $\varepsilon$ increased. The highest average test accuracy of F-GR was better than those of B-GR and DB although our purpose is to confirm the difference among the algorithms and not to achieve higher accuracy by tuning both $\gamma$ and $\varepsilon$. It was $(\text{F-GR}, \text{B-GR}, \text{DB}) = (58.6, 58.3, 57.6) \pm (0.2, 0.2, 0.2)$ for MLP and $(87.0, 86.2, 86.3) \pm (0.2, 0.3, 0.3)$ for ResNet-18. We also confirmed that the same tendencies appeared in the grid search of ResNet-34 on CIFAR-100 (Figure S.3). Moreover, we confirmed that F-GR performed better than B-GR and DB in a more realistic training of a wide residual network (WRN-28-10) on CIFAR-10 and CIFAR-100 with/without data augmentation (Table S.1).

It is noteworthy that the best accuracy of F-GR was obtained close to the line of $\gamma = \varepsilon$. This line is closely related to SAM algorithm. We explain more details in Section 5.1. We also observed that when the ascent step was too small (e.g., $\varepsilon \lesssim 10^{-4}$), numerical instability sometime appeared in the calculation of the finite difference $\Delta R$. Overall, the experiments suggest that F-GR with a large ascent step is better to use for achieving higher generalization performance.

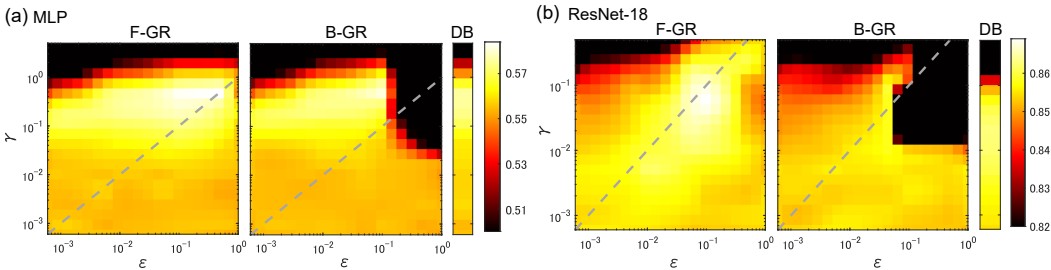

Figure 3: Grid search on learning with different GR algorithms shows the superiority of F-GR and that a relatively large $\varepsilon$ achieves a high test accuracy. The color bar shows the average test accuracy over 5 trials. Gray dashed lines indicate $\gamma = \varepsilon$.

## 4 THEORETICAL ANALYSIS OF IMPLICIT BIAS

Although previous work and our experiments in Section 3.3 indicate improvements of prediction performance caused by GR, theoretical understanding of this phenomenon remains limited. Because the gradient norm itself eventually becomes zero after the model achieves a zero training loss, it seems challenging to distinguish the generalization capacity by simply observing the value of the gradient norm after training. In addition, our experiments clarified that the performance also depends on the choice of the algorithm and revealed that the situation is complicated. To attack this problem, we consider a solvable model and reveal that GR methods actually contribute to the selection of global minima and the eventual performance.

### 4.1 DIAGONAL LINEAR NETWORK

A DLN is a solvable model proposed by Woodworth et al. (2020). It is a linear transformation of input $x \in \mathbb{R}^d$ defined as $\langle \beta, x \rangle$ where $\beta$ is parameterized by $\beta = w_+^2 - w_-^2$ with $w = (w_+, w_-) \in \mathbb{R}^{2d}$. Here, the square of the vector is an element-wise square operation. Suppose we have $n$ training samples $(x_i, y_i)$ $(i = 1, ..., n)$. The training loss is given by

$$\mathcal{L}(w) = \frac{1}{4n} \sum_{i=1}^{n} \left( \langle w_+^2 - w_-^2, x_i \rangle - y_i \right)^2. \tag{8}$$

Consider continual-time training dynamics $dw/dt = -\nabla \mathcal{L}$. We set an initialization $w_+(t = 0) = w_-(t = 0) = \alpha_0$ which is a $d$-dimensional vector and whose entries are non-zero. We define a data matrix $X$ whose $i$-th row is given by $x_i$. Woodworth et al. (2020) found that interpolation solutions of usual gradient descent are given by

$$\beta_\infty(\alpha) = \underset{\beta \in \mathbb{R}^d \text{ s.t. } X\beta = y}{\arg \min} \phi_\alpha(\beta) \tag{9}$$

with $\alpha = \alpha_0$. The potential function $\phi_\alpha$ is given by $\phi_\alpha(\beta) = \sum_{i=1}^{d} \alpha_i^2 q \left( \beta_i / \alpha_i^2 \right)$ with $q(z) = 2 - \sqrt{4 + z^2} + z \operatorname{arcsinh}(z/2)$. For a larger scale of initialization $\alpha$, this potential function becomes closer to L2 regularization as $\alpha_i^2 q(\beta_i / \alpha_i^2) \sim |\beta_i|^2$, which corresponds to the L2 min-norm solution of the lazy regime (Chizat et al., 2019). In contrast, for a smaller scale of initialization $\alpha$, it becomes closer to L1 regularization as $\alpha_i^2 q(\beta_i / \alpha_i^2) \sim |\beta_i|$. In this way, we can observe a one-parameter interpolation between L1 and L2 implicit biases. Deep neural networks in practice acquire rich features depending on data structure and are believed to be beyond the lazy regime. Thus, obtaining an L1 solution by setting small $\alpha$ is referred to as the *rich regime* and desirable. Previous work has revealed that effective values of $\alpha$ decreases by a larger learning rate in the discrete update (Nacson et al., 2022), SGD (Pesme et al., 2021), and SAM update (Andriushchenko & Flammarion, 2022). These learning methods have an implicit bias that chooses the L1 sparse solution in the rich regime.

### 4.2 IMPLICIT BIAS OF GR

Now, consider gradient descent with F-GR $dw/dt = -\nabla \mathcal{L}(w) - \gamma \Delta R_F(w)$. We find that the GR has implicit bias for the rich regime, and moreover, the strength of the bias depends on the ascent step size.

**Theorem 4.1.** *Assume that (i) the gradient dynamics converges to the interpolation solution satisfying $X\beta = y$, (ii) L2 norm of the parameter $\|w(t)\|$ has a constant upper bound independent of $\gamma$ and $\varepsilon$, (iii) for sufficiently small $\gamma$ and $\varepsilon$, the integral of the training loss, i.e., $\int_0^\infty \mathcal{L}(w(t))dt$, has a constant upper (lower, respectively) bound $\overline{R}$ ($\underline{R}$) independent of $\gamma$ and $\varepsilon$. Then, for sufficiently small $\gamma$ and $\varepsilon$, interpolation solutions are given by $\beta_\infty(\alpha_{F\text{-}GR})$ with*

$$\alpha_{F\text{-}GR} \le \alpha_0 \circ \exp(-\gamma \varepsilon c^* + \mathcal{O}(\gamma^2) + \mathcal{O}(\varepsilon^2)). \tag{10}$$

*The exponent $c^* \in \mathbb{R}^d$ is a non-negative constant vector given by*

$$c^* = \frac{1}{2n^2} (X^\top (X\beta(t = 0) - y))^2. \tag{11}$$

Note that the inequality is element-wise. The proof is given in Section A.1. Technically speaking, learning with F-GR requires to evaluate a novel $c^*$ term, which has not appeared in the analyses of

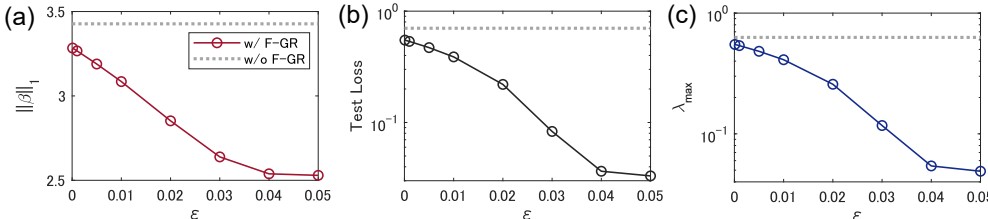

Figure 4: Results of training of DLNs using gradient descent with F-GR ($\gamma = 0.02$). (a) L1 norm of the solutions, (b) test loss, and (c) the largest eigenvalue of the Hessian of the training loss.

previous studies. Lemma A.1 clarifies that we can prove the positivity of the seemingly complicated term of $c^*$ through an integral of the learning dynamics. The assumptions seem rational in the following sense. First, assumption (i) is common in the previous studies on DLNs. Second, Nacson et al. (2022) recently reported that we can obtain interpolation solutions with a smaller parameter norm $\|w(t)\|$ using the discrete update with a larger learning rate. Because the interpolation solutions of gradient descent are also those of our learning with GR, assumption (ii) seems rational. The upper bound of assumption (iii) means that the convergence speed of $\mathcal{L}(w(t))$ does not get too small for sufficiently small $\gamma$ and $\varepsilon$. As a side note, we can replace assumption (iii) with the positive definiteness of a certain matrix (assumption A.2). This is seemingly rather technical, but related to a sufficient condition that the dynamics converge to the global minima. See Section A.2 for details.

This theorem reveals that GR has an implicit bias to select the L1 solution, that is, the rich regime because $\alpha$ is always smaller than $\alpha_0$. As the ascent step increases, we have an exponentially smaller upper bound and the implicit bias to L1 solution will become stronger. We confirm this dependence of solutions on the ascent step in numerical experiments (Figure 4). As in previous work, we trained DLNs on the synthetic data of a sparse regression problem, where $x_i \sim \mathcal{N}(\mu 1, \sigma^2 I)$ and $y_i \sim \mathcal{N}(\langle \beta^*, x_i \rangle, 0.01)$, and where $\beta^*$ is $k^*$-sparse with non-zero entries equal to $1/\sqrt{k^*}$ ($d = 100$ and $n = 50$). Following Nacson et al. (2022), we chose $\mu = \sigma^2 = 5$, where the parameter norm $a(t)$ is suppressed and assumption (ii) ix expected to hold. As the ascent step increases, models trained by F-GR obtain sparser solutions (Figure 4(a)) and better generalization performance (Figure 4(b)). The dashed lines show the results of gradient descent without GR. This result is consistent with our experiments of in more realistic settings (Figure 3) where a relatively large $\varepsilon$ achieves the best performance. In Figure 4(c), we also present the largest eigenvalue of the Hessian (S.50), computed after training. As the ascent step size increases, F-GR chooses flatter minima. This is also consistent with empirical observations in previous studies on GR. Note that B-GR can potentially make the bound looser as the step size $\varepsilon$ increases since B-GR is equivalent to F-GR with $-\varepsilon$. Actually, we can immediately find a lower bound $\alpha_{B\text{-}GR} \gtrsim C \circ \exp(\gamma \varepsilon c^*)$ for a positive constant vector $C$, as is remarked in Section A.1. The results of numerical experiments on DLNs shown in Figure S.3 confirm that learning with F-GR achieved better generalization performance than B-GR.

While Theorem 4.1 gives us insight into the finite-difference GR, the upper bound converges to $\alpha_0$ for the DB limit ($\varepsilon \to 0^+$) and becomes meaningless. Fortunately, we can construct an upper bound applicable to the DB limit.

**Proposition 4.2.** *Under the same assumptions as in Theorem 4.1, for sufficiently small $\varepsilon$ and $\gamma$,*

$$\alpha_{F\text{-}GR} \leq \alpha_0 \circ \exp(-\gamma c + \mathcal{O}(\gamma^2) + \mathcal{O}(\varepsilon^2)), \tag{12}$$

*where the exponent $c \in \mathbb{R}^d$ is a non-negative variable given by $c = n^{-2} \int_0^\infty (X^\top (X\beta(s) - y))^2 ds$.*

Its derivation is given in Section A.3. One can regard this proposition as a minor extension of Theorem 1 in Andriushchenko & Flammarion (2022), which has investigated $\gamma = \varepsilon$. This setting has a special meaning as we mention in Section 5.1. From the proposition, one can see that the DB limit still has the implicit bias to select the rich regime. This is consistent with the numerical experiments in Figure 4 where the limit of small $\varepsilon$ achieves slightly better and sparser solutions than GD without GR. Although the bound (12) is informative, it is difficult to evaluate a concrete value of $c$. As a side note, we can make a bound of the average over entries, that is, $\sum_{i=1}^d c_i/d \geq (4n/d)\lambda_{min}(XX^\top)\underline{R}$. See Section A.3 for details.

## 5 GR IN GRADIENT-ASCENT-AND-DESCENT LEARNING

We have revealed that learning with finite-difference GR, F-GR in particular, improves performance. We recall that the GR is composed of both gradient ascent and descent steps. This computation makes the GR essentially related to two other learning methods similarly composed of both gradient ascent and descent steps: the SAM algorithm and the flooding method.

### 5.1 CONNECTION WITH SAM

The SAM algorithm was derived from the minimization of a surrogate loss $\max_{\|\varepsilon\| \leq \rho} \mathcal{L}(\theta + \varepsilon)$ for a fixed $\rho > 0$, and has achieved the highest performance in various models (Foret et al., 2021). After some heuristic approximations, its update rule reduces to iterative gradient ascent and descent steps: $\theta_{t+1} = \theta_t - \eta \nabla \mathcal{L}(\theta')$ with $\theta' = \theta_t + \varepsilon_t \nabla \mathcal{L}(\theta_t)$ and $\varepsilon_t = \rho / \|\nabla \mathcal{L}(\theta_t)\|$. Under a specific condition, the SAM update can be seen as gradient descent with F-GR. Let us consider time-dependent regularization coefficient $\gamma_t$ and ascent step $\varepsilon_t$. Then, for $\gamma_t = \varepsilon_t$, the gradient descent with F-GR becomes equivalent to the SAM update:

$$\nabla \mathcal{L}(\theta) + \frac{\gamma_t}{\varepsilon_t}(\nabla \mathcal{L}(\theta') - \nabla \mathcal{L}(\theta)) = \nabla \mathcal{L}(\theta'). \tag{13}$$

A similar equivalence has been pointed out in Zhao et al. (2022) which supposes a non-squared gradient norm and $\varepsilon_t = \rho / \|\nabla \mathcal{L}(\theta_t)\|$ naturally appears. Let us focus on the SAM update without the gradient normalization for simplicity, that is, $\varepsilon_t = \rho$. This simplified SAM update was analyzed on DLNs in Andriushchenko & Flammarion (2022). We can recover the SAM case by setting a sufficiently small $\gamma = \varepsilon$ in Proposition 4.2. Although it will be curious to identify any optimal setting of $(\gamma, \varepsilon)$, our analysis is limited to the range of the first-order Taylor expansion and characterizing any optimal setting seems beyond the scope of our analysis. In Figure 3, we empirically observed the optimal setting for generalization was very close to or just on the line $\gamma = \varepsilon$. In contrast, our Figures 4, S.3 and the previous study Zhao et al. (2022) demonstrated that the optimal setting was not necessarily on $\gamma = \varepsilon$, and thus combining the ascent and descent steps would be still promising.

### 5.2 FLOODING PERFORMS GR IN AN IMPLICIT WAY

The flooding method (Ishida et al., 2020) is another learning algorithm composed of both gradient ascent and descent steps. Its update rule is given by

$$\theta_{t+1} = \theta_t - \eta \operatorname{Sign}(\mathcal{L} - b) \nabla \mathcal{L} \tag{14}$$

for a constant $b > 0$, referred to as the flood level. When the training loss becomes lower than the flood level, the sign of the gradient is flipped and the parameter is updated by gradient ascent. Therefore, the flooding causes the training dynamics to continue to wander around $\mathcal{L}(\theta) \sim b$, and its gradient continues to take a non-zero value. This would seem a kind of early stopping, but previous work empirically demonstrates that flooding performs better than naive early stopping and finds flat minima. For simplicity, let us focus on the gradient descent for a full batch. The following theorem clarifies a hidden mechanism of flooding.

**Theorem 5.1.** *Consider the time step $t$ satisfying $\mathcal{L}(\theta_t) < b$ and $\mathcal{L}(\theta_{t+1}) > b$. Then, the flooding update from $\theta_t$ to $\theta_{t+2}$ is equivalent to the gradient of the F-GR with $\varepsilon = \gamma = \eta$:*

$$\theta_{t+2} = \theta_t - \eta^2 \frac{\nabla \mathcal{L}(\theta_t + \eta \nabla \mathcal{L}(\theta_t)) - \nabla \mathcal{L}(\theta_t)}{\eta}. \tag{15}$$

*Similarly, for $\mathcal{L}(\theta_t) > b$ and $\mathcal{L}(\theta_{t+1}) < b$, the flooding update is equivalent to the gradient of the B-GR.*

Although its derivation is quite straightforward (see Section B), this essential connection between finite-difference GR and flooding has been missed in the literature. Ishida et al. (2020) conjectured that flooding causes a random walk on the loss surface and this would contribute to the search for flat minima in some ways. Our result implies that the dynamics of flooding are not necessarily random and it can actively search the loss surface in a direction that decreases the GR. This is consistent with the observations that the usual gradient descent with GR finds flat minima (Barrett & Dherin, 2021; Zhao et al., 2022). Note that the ascent step is given by the learning rate $\eta$, and $\eta$ is usually decayed

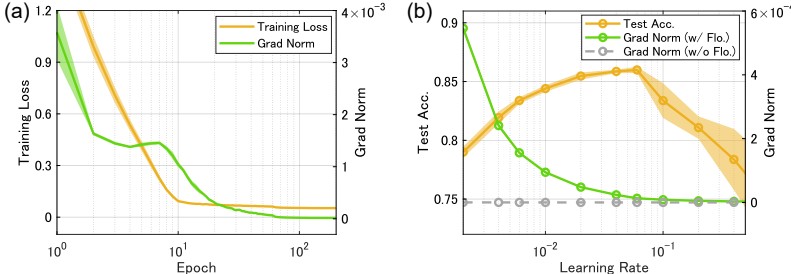

Figure 5: Flooding decreases the gradient norm, as expected by theory. (a) Training dynamics of flooding with $b = 0.05$. (b) Test accuracy and gradient norm after the training.

in the training. This implies that because the ascent step size is relatively small, the implicit B-GR in the flooding update would not make the generalization performance much worse.

Figure 5 empirically confirms that the flooding method decreases the gradient norm $R(\theta)$. We trained ResNet-18 on CIFAR-10 by using flooding. Figure 5(a) shows that at the beginning of the training, the training loss decreases in the usual way because the loss is far above flood level $b$. Around the 10th epoch, the loss value becomes sufficiently close to the flood level for the decrease in the loss to slow (Figure S.5 ). Then, the flooding update becomes dominant in the dynamics the gradient norm begins to decrease. Figure 5(b) demonstrates that the gradient norm of the trained model decreases as the initial learning rate increases. This is consistent with Theorem 5.1 because the theorem claims that the larger learning rate induces the larger regularization coefficient of the GR $\gamma = \eta$. In contrast, naive SGD training without flooding always reaches an almost zero gradient norm regardless of the learning rate. Thus, the change in the gradient norm depending on the learning rate is specific to flooding and implies that it implicitly performs GR.

## 6 DISCUSSION

This work presented novel practical and theoretical insights into GR. The finite-difference computation is effective in the sense of both reducing computational cost and improving performance. Theoretical analysis supports the empirical observation that the forward difference computation has an implicit bias that chooses potentially better minima depending on the size of the ascent step. Because deep learning requires large-scale models, it would be reasonable to use learning methods only composed of first-order descent or ascent gradients. The current work suggests that the F-GR is a promising direction for further investigation and could be extended for our understanding and practical usage of gradient-based regularization.

We suggest several potentially interesting research directions. From a broader perspective, we may regard finite-difference GR, SAM, and flooding as a single learning framework composed of iterative gradient ascent and descent steps. It would be interesting to investigate if there is optimal combination of these steps for further improving performance. As our experiments suggest, only using the gradient descent or ascent does not necessarily achieve the best performance, and a combination of them seems to be the best approach. Similar results were empirically observed in other gradient-based regularization techniques (Zhao et al., 2022; Zhuang et al., 2022). Related to the combination between the gradient descent and ascent, although we fixed the ascent step size as a constant, a step size decay or any scheduling could enhance the performance further. For instance, Zhuang et al. (2022) used a time-step dependent ascent step to achieve high prediction performance for SAM.

It will also be interesting to explore any theoretical clarification beyond the scope of DLNs. Although a series of analyses in DLNs enable us to explore the implicit bias for selecting global minima, it assumes global convergence and avoids an explicit evaluation of convergence dynamics. Thus, it would be informative to explore the convergence rate or escape from local minima in other solvable models or a more general formulation if possible. Constructing generalization bounds would also be an interesting direction. Some theoretical work has proved that regularizing first-order derivatives of the network output control the generalization capacity (Ma & Ying, 2021), and such derivatives are included in the gradient norm as a part. We expect that the current work will serve as a foundation for further developing and understanding regularization methods in deep learning.

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

# Appendices

## A    ANALYSIS BY DIAGONAL LINEAR NETWORKS

### A.1    PROOF OF THEOREM 4.1

#### A.1.1    INTERPOLATION SOLUTIONS BETWEEN L1 AND L2 REGULARIZATION

We consider the training dynamics with F-GR as

$$\dot{w}_t = -\nabla\mathcal{L}(w_t) - \gamma\frac{\nabla\mathcal{L}(w_t + \varepsilon\nabla\mathcal{L}(w_t)) - \nabla\mathcal{L}(w_t)}{\varepsilon} \tag{S.1}$$

$$= -q_1\nabla\mathcal{L}(w_t) - q_2\nabla\mathcal{L}(w_t + \varepsilon\nabla\mathcal{L}(w_t)), \tag{S.2}$$

where $q_1 = (1 - \gamma/\varepsilon)$, $q_2 = \gamma/\varepsilon$. The training loss $\mathcal{L}(w)$ is defined in (8). The dynamics are rewritten as

$$\frac{dw(t)}{dt} = -\frac{q_1}{n}(\tilde{X}^\top r(t)) \circ w(t) - \frac{q_2}{n}(\tilde{X}^\top r^*(t)) \circ w^*(t), \tag{S.3}$$

where $\circ$ denotes the element-wise product between vectors. We defined $r(t) = \tilde{X}w(t)^2 - y$, $r^*(t) = \tilde{X}w^*(t)^2 - y$, $w^*(t) = w(t) + \varepsilon\nabla\mathcal{L}(w(t))$, and put $\tilde{X} = [\ X\ \ -X\ ] \in \mathbb{R}^{n\times 2d}$. We recall that the square of the vector is an element-wise square operation. The general solution of (S.3) is written as

$$w(t) = \begin{bmatrix} \alpha_0 \\ \alpha_0 \end{bmatrix} \circ \exp\left(-\frac{1}{n}\tilde{X}^\top \int_0^t (q_1 r(s) + q_2 r^*(s))ds\right)$$

$$\circ \exp\left(-\frac{q_2\varepsilon}{n^2}\int_0^t (\tilde{X}^\top r^*(s)) \circ (\tilde{X}^\top r(s))ds\right). \tag{S.4}$$

This recovers the GD solution obtained by Woodworth et al. (2020) for $(q_1, q_2) = (1, 0)$, and SAM solution by Andriushchenko & Flammarion (2022) for $(q_1, q_2) = (0, 1)$. To evaluate the effect of both $\varepsilon$ and $\gamma$ on the implicit bias, we need a lower-order evaluation compared to previous work.

Suppose an interpolation solution $\beta_\infty$ satisfying $X\beta_\infty = y$. We can represent it by

$$\beta_\infty = w_+(\infty)^2 - w_-(\infty)^2 \tag{S.5}$$

$$= 2\alpha_{F\text{-}GR}^2 \circ \sinh\left(X^\top\nu\right), \tag{S.6}$$

where $\nu = -\frac{2}{n}\int_0^\infty (q_1 r(s) + q_2 r^*(s))ds$ and

$$\alpha_{F\text{-}GR} := \alpha_0 \circ \exp\left(-\frac{\gamma}{n^2}\Psi\right), \quad \Psi := \int_0^\infty \left(X^\top r^*(s)\right) \circ \left(X^\top r(s)\right) ds. \tag{S.7}$$

Put $\beta_\infty = B_{\alpha_{F\text{-}GR}}\left(X^\top\nu\right)$ with $B_{\alpha_{F\text{-}GR}}(z) = 2\alpha_{F\text{-}GR}^2 \circ \sinh(z)$. Because the form of the function $\beta_\infty = B_\alpha\left(X^\top\nu\right)$ is the same as in the analysis of usual gradient descent (Woodworth et al., 2020), we can use their transformation of $\beta_\infty$ as it is. We have a KKT condition $\nabla\phi_\alpha(w) = X^\top\nu$ and the function $\phi_\alpha$ satisfies

$$\nabla\phi_\alpha(\beta) = B_\alpha^{-1}(\beta) = \operatorname{arcsinh}\left(\frac{1}{2\alpha^2}\circ\beta\right). \tag{S.8}$$

We have

$$\beta_\infty(\alpha) = \underset{\beta\in\mathbb{R}^d \text{ s.t. } X\beta=y}{\arg\min}\ \phi_\alpha(\beta) \tag{S.9}$$

with

$$\phi_\alpha(\beta) = \sum_{i=1}^d \alpha_i^2 q\left(\beta_i/\alpha_i^2\right) \tag{S.10}$$

and

$$q(z) = 2 - \sqrt{4 + z^2} + z\operatorname{arcsinh}(z/2). \tag{S.11}$$

In our GR case, $\alpha$ is just replaced by $\alpha_{F\text{-}GR}$.

### A.1.2 Evaluation on $\alpha_{F\text{-}GR}$

From the definitions of $r(t)$ and $r^*(t)$, we have

$$r^*(t) - r(t) = \frac{2\varepsilon}{n}\tilde{X}((\tilde{X}^\top r(t)) \circ w(t)^2) + \frac{\varepsilon^2}{n^2}\tilde{X}((\tilde{X}^\top r(t))^2 \circ w(t)^2). \tag{S.12}$$

Then,

$$\Psi = \int_0^\infty (X^\top r(s))^2 ds + \frac{\varepsilon}{n} \int_0^\infty \underbrace{2(X^\top \tilde{X}((\tilde{X}^\top r(s)) \circ w(s)^2)) \circ (X^\top r(s))}_{=:z(s)} ds$$

$$+ \frac{\varepsilon^2}{n^2} \int_0^\infty \underbrace{(X^\top \tilde{X}((\tilde{X}^\top r(s))^2 \circ w(t)^2)) \circ (X^\top r(s))}_{=:z_h(s)} ds. \tag{S.13}$$

Let us put

$$\Psi = \Psi_0 + \frac{\varepsilon}{n}\Psi_1 + \frac{\varepsilon^2}{n^2}\Psi_2, \tag{S.14}$$

$$\Psi_0 := \int_0^\infty (X^\top r(s))^2 ds, \quad \Psi_1 := \int_0^\infty z(s)ds, \quad \Psi_2 := \int_0^\infty z_h(s)ds. \tag{S.15}$$

Note that the first term $\Psi_0$ essentially corresponds to the implicit bias of the SAM update investigated in the previous study (Andriushchenko & Flammarion, 2022). Because the SAM update corresponds to $\gamma = \varepsilon$, the dominant term of $\gamma\Psi$ is $\Psi_0$ and they neglect the other terms. In our GR case, $\gamma$ and $\varepsilon$ have different scales in general and we need to evaluate the coefficient of the ascent step, that is, $\Psi_1$.

**Lemma A.1.** *Under assumptions (i)-(iii), for sufficiently small $\gamma$, $\Psi_1 > nb(0)^2/2 + \mathcal{O}(\gamma)$. If we further assume $b_i(0) \neq 0$ for all $i$, $\Psi_1 > nb(0)^2/4$.*

*Proof of Lemma A.1.* The dynamics (S.3) are rewritten as

$$n\frac{dw}{dt} = -\tilde{b} \circ w - \frac{\gamma}{n}[2(\tilde{Z}(\tilde{b} \circ w^2)) \circ w + \tilde{b}^2 \circ w]$$

$$- \frac{\gamma\varepsilon}{n^2}[(\tilde{Z}(\tilde{b}^2 \circ w^2)) \circ w + 2(\tilde{Z}(\tilde{b} \circ w^2)) \circ w \circ \tilde{b}] - \frac{\gamma\varepsilon^2}{n^3}[(\tilde{Z}(\tilde{b}^2 \circ w^2)) \circ w \circ \tilde{b}], \tag{S.16}$$

where we put $\tilde{b} = \tilde{X}^\top r$ and $\tilde{Z} = \tilde{X}^\top \tilde{X}$. This gives us

$$\frac{n}{2}\frac{d\beta}{dt} = -b \circ a - \frac{\gamma}{n}\underbrace{[2(Z(b \circ a)) \circ a + b^2 \circ \beta]}_{=:Q_1(t)}$$

$$- \frac{\gamma\varepsilon}{n^2}\underbrace{[(Z(b^2 \circ \beta)) \circ a + 2(Z(b \circ a)) \circ \beta \circ b]}_{=:Q_2(t)} - \frac{\gamma\varepsilon^2}{n^3}\underbrace{[(Z(b^2 \circ \beta)) \circ \beta \circ b]}_{=:Q_3(t)}, \tag{S.17}$$

where we put $a = w_+^2 + w_-^2$, $b = X^\top r$ and $Z = X^\top X$. Note that $db/dt = X^\top(dr/dt) = X^\top X(d\beta/dt)$. By multiplying $X^\top X$ to (S.17) and taking the Hadamard product with $b$, we have

$$n\frac{db^2}{dt} = -4b \circ (X^\top X(b \circ a)) - \frac{4\gamma}{n}\underbrace{b \circ [X^\top X(Q_1(t) + \frac{\varepsilon}{n}Q_2(t) + \frac{\varepsilon^2}{n^2}Q_3(t))]}_{=:Q(t)}. \tag{S.18}$$

The point is that we have $2b \circ (X^\top X(b \circ a)) = z(t)$. This relation makes us to evaluate the seemingly complicated term $\Psi_1$ by the change of $b(t)^2$, which corresponds to a training loss. By taking the integral over time, the above dynamics become

$$\Psi_1 = \int_0^\infty z(s)ds = \frac{n}{2}b(0)^2 - 2\frac{\gamma}{n}\int_0^\infty Q(s)ds. \tag{S.19}$$

We used assumption (i) that we have a global minimum and $b(\infty) = 0$. If $\gamma$ is sufficiently small and $\int_0^\infty Q(s)ds$ is finite, we will have a non-negative $\Psi_1$.

Here, we use assumption (ii) that the parameter norm has a finite constant upper bound independent of $\gamma$ and $\varepsilon$. Because $\|a(t)\| = \|w_+(t)^2 + w_-(t)^2\| \leq \|w\|^2$, we have an upper bound of $\|a(t)\|$ as well:

$$\|a(t)\| \leq \bar{a}. \tag{S.20}$$

Define $\kappa_1 := \arg\max_i \|Xx_i\|$, $\kappa_2 := \arg\max_i \|x_i\|$ and $\kappa_3 := \|XX^\top\|_2$. Then, we find

$$|Q_{1,i}(t)| \leq 2a_i\|Xx_i\|\|b \circ a\| + b_i^2|\beta_i| \tag{S.21}$$

$$\leq 2\bar{a}^2\kappa_1\sqrt{\kappa_3}\|r(t)\| + \bar{a}\kappa_2^2\|r(t)\|^2. \tag{S.22}$$

where we used $\|b \circ a\| \leq \|b\|\|a\| \leq \sqrt{\kappa_3}\bar{a}\|r\|$ and $\|\beta\| \leq \|a\| \leq \bar{a}$. Similarly, we have

$$|Q_{2,i}(t)| \leq \bar{a}^2\kappa_1\kappa_3\|r\|^2 + 2\bar{a}^2\kappa_1\kappa_2\sqrt{\kappa_3}\|r\|^2, \tag{S.23}$$

where we used $\|b^2\| \leq \sqrt{\sum_i (X_i r)^4} \leq \sum_i (X_i r)^2 = \|b\|^2$. We also have

$$|Q_{3,i}(t)| \leq \bar{a}^2\kappa_1\kappa_2\kappa_3\|r\|^3. \tag{S.24}$$

Note that under assumption (ii), the training loss is upper-bounded as well because

$$\|r(t)\| \leq \|X\beta\| + \|y\| \leq \sqrt{\kappa_3}\bar{a} + \|y\| =: \bar{\mathcal{L}}. \tag{S.25}$$

Therefore, we have

$$|Q_{3,i}(t)| \leq \|a\|\kappa_1\kappa_2\kappa_3\bar{\mathcal{L}}\|r\|^2. \tag{S.26}$$

After all, the inequalities (S.22 ,S.23 ,S.26 ) lead to

$$\int_0^\infty ds\, Q_i(s) \leq C \int_0^\infty ds\|r\|^2, \tag{S.27}$$

where $C$ denotes an uninteresting positive constant. By using assumption (iii) that $\int_0^\infty ds\|r\|^2$ has an constant upper bound $4n\bar{R}$ independent of $\gamma$ and $\varepsilon$, we have

$$\Psi_1 \geq \frac{nb(0)^2}{2} - 8\gamma C\bar{R}. \tag{S.28}$$

Therefore, for sufficiently small $\gamma$, the dominant term is non-negative. Moreover, if we have $b_i(0) \neq 0$ for all $i$,

$$\Psi_1 \geq \frac{nb(0)^2}{4} > 0 \ \ \text{for} \ \ \gamma < \min_i \frac{nb_i(0)^2}{32C\bar{R}}. \tag{S.29}$$

$\blacksquare$

As a side note, the inequality (S.29 ) of $\gamma$ gives us some insight into non-asymptotic evaluation on how large $\gamma$ we can take. First, the constant $C$ includes $\bar{a}$ and it implies that we need a smaller $\gamma$ for a larger parameter norm $\bar{a}$. Second, note that $\bar{R}$ controls the integral of the training loss over the whole training dynamics. We need a smaller $\gamma$ as well for a larger $\bar{R}$ which implies the convergence of dynamics is slower.

Next, we evaluate $\Psi_2$. Since

$$z_h(s) = (Z(b^2 \circ \beta)) \circ b, \tag{S.30}$$

we have

$$|z_{h,i}| \leq \kappa_1\kappa_2\kappa_3\bar{a}\bar{L}\|r\|^2. \tag{S.31}$$

Therefore,

$$\left| \int_0^\infty z_{h,i}(s)ds \right| \leq C'\bar{R}. \tag{S.32}$$

exThus, $\Psi_2$ is finite and becomes negligible in $\Psi$ for a sufficiently small $\varepsilon$.

Finally, we have

$$\gamma\Psi \geq \varepsilon\gamma\frac{b(0)^2}{2} - \frac{8\varepsilon\gamma^2}{n}C\bar{R} - \frac{\varepsilon^2\gamma}{n^2}C'\bar{R}, \tag{S.33}$$

where we used Lemma A.1. Substituting the above inequality and $b(0) = X^\top r(0)$ into (S.7), we obtain Theorem 4.1.

**Remark on higher-order terms in Theorem 4.1**: First, let us remark on $\mathcal{O}(\gamma^2)$ term. Lemma A.1 tells us that if we have $b_i(0) \neq 0$ for all $i$, we have a slightly stronger result than Theorem 4.1:

$$\alpha_{F\text{-}GR} \leq \alpha_0 \circ \exp(-\gamma\varepsilon c^*/2 + \mathcal{O}(\varepsilon^2)), \tag{S.34}$$

where the $\mathcal{O}(\gamma^2)$ term disappears. The condition of $b_i(0) \neq 0$ seems to hold in usual cases because the network parameters and training samples are randomly assigned at initialization. Second, regarding $\mathcal{O}(\varepsilon^2)$ term, we have $\Psi \geq 0$ for

$$\varepsilon \leq \frac{n^2}{C'\bar{R}} \left( \min_i \frac{b_i(0)^2}{2} - \frac{8\gamma}{n} C\bar{R} \right) \tag{S.35}$$

$$< \min_i \frac{n^2 b_i(0)^2}{4C'\bar{R}} \tag{S.36}$$

where the first line comes from (S.33) and the second one from a small $\gamma$ satisfying (S.33). This implies that we need a smaller $\varepsilon$ for larger $\bar{a}$ and $\bar{R}$ in a similar way to $\gamma$.

**Remark on B-GR:** We have obtained the upper bound of $\alpha_{F\text{-}GR}$, that is, the lower bound of $\Psi$ for F-GR. Since we can see B-GR as the F-GR with a negative $\varepsilon$, the sign of $\Psi_1$ is flipped in B-GR. Then, we can easily obtain the upper bound of $\Psi$ as follows.

First, we have

$$\Psi_0 \leq 2\lambda_{max}(XX^\top) \int_0^\infty \|r(s)\|^2 ds \tag{S.37}$$

$$\leq 8n\lambda_{max}(XX^\top)\bar{R}, \tag{S.38}$$

where $\lambda_{max}(XX^\top)$ denotes the largest eigenvalue of $XX^\top$. We have

$$\Psi = \Psi_0 - \frac{\varepsilon}{n}\Psi_1 + \mathcal{O}(\varepsilon^2) \tag{S.39}$$

$$\lesssim 8n\lambda_{max}(XX^\top)\bar{R} - \varepsilon\frac{b(0)^2}{2}, \tag{S.40}$$

where we used Lemma A.1. Substituting the above inequality into (S.7), we obtain $\alpha_{F\text{-}GR} \gtrsim C \circ \exp(\gamma\varepsilon c^*)$ for a positive constant $C$. Thus, the lower bound increases for a larger step size $\varepsilon > 0$ in B-GR and the implicit bias is strengthen in the direction to L2 solutions.

## A.2 ALTERNATIVE TO ASSUMPTION (III)

Instead of assumption (iii), we may use

**Assumption A.2.** *For sufficiently small $\varepsilon$ and $\gamma$, the smallest eigenvalue of $S(t) := X \, diag(a(t))X^\top$ is positive.*

Since we suppose the overparameterized case ($d > n$), the matrix $X$ is a wide matrix and $S$ has no trivial zero eigenvalue. The positive definiteness of $S$ is a sufficient condition of global convergence as follows. From Eq. (S.17), we have

$$\frac{n}{4}\frac{d\|r\|^2}{dt} = \frac{n}{2}b^\top\frac{d\beta}{dt} = -r^\top Sr - \frac{\gamma}{n}r^\top X(Q_1(t) + \frac{\varepsilon}{n}Q_2(t) + \frac{\varepsilon^2}{n^2}Q_3(t)). \tag{S.41}$$

Using the inequalities (S.22, S.23, S.26), we have

$$\frac{n}{4}\frac{d\|r\|^2}{dt} \leq -\lambda_{min}^*\|r\|^2 + \gamma C\|r\|^2. \tag{S.42}$$

where we take the lower bound of the smallest eigenvalue as $\lambda_{min}^* = \min_{t,\gamma,\varepsilon} \lambda_{min}(S(t))$. By taking a sufficiently small $\gamma$ such that $\gamma < 3\lambda_{min}^*/(4C)$, we obtain

$$\|r(t)\|^2 \leq \|r(0)\|^2 \exp(-\lambda_{min}^* t/n), \tag{S.43}$$

from Grönwall's inequality. Since $\mathcal{L}(w(t)) = \|r(t)\|^2/(4n)$, we obtain global convergence. In addition, we have

$$\int_0^\infty ds \|r(s)\|^2 \le \|r(0)\|^2 \int_0^\infty ds \exp(-\lambda_{min}^* t/n) = n\|r(0)\|^2/\lambda_{min}^*. \tag{S.44}$$

This gives the upper bound $\bar{R}$. Similarly, we obtain $\underline{R}$ by taking the lower bound of (S.41) and using Grönwall's inequality. Thus, instead of assumption (iii), we can apply Assumption A.2 in the transformation from (S.27) to (S.29).

Note that $S(t)$ is known as the neural tangent kernel in the lazy regime and its positive definiteness is straightforward (Woodworth et al., 2020). Although there is no proof of the positive definiteness in the rich regime, we observed it in numerical experiments and the assumption A.2 seems rational.

### A.3 DERIVATION OF PROPOSITION 4.2

We obtained the upper bound of $\alpha_{F\text{-}GR}$ (in other words, the lower bound of $\Psi$) from the term of $\Psi_1$. In some cases, $\Psi_0$ gives us complementary insight.

**Proposition A.3.** *Under the same assumptions as in Theorem 4.1, for sufficiently small $\varepsilon$ and $\gamma$,*

$$\alpha_{F\text{-}GR} \le \alpha_0 \circ \exp(-\gamma c + \mathcal{O}(\gamma^2) + \mathcal{O}(\varepsilon^2)), \tag{S.45}$$

*where the exponent $c$ is a non-negative variable given by $c = n^{-2}\int_0^\infty (X^\top(X\beta(s) - y))^2 ds$.*

*Proof.* $\Psi_0$ is non-negative by definition and written as

$$\Psi_0 = \int_0^\infty (X^\top(X\beta(s) - y))^2 ds. \tag{S.46}$$

In addition, we have $\Psi_1 \ge \mathcal{O}(\gamma)$ from Lemma A.1. Therefore, we have $\Psi \ge \Psi_0 + \mathcal{O}(\gamma) + \mathcal{O}(\varepsilon^2)$. Substituting this into (S.7), we obtain the result. $\qquad\square$

**Remark on an average of $c$:** Note that $c$ may depend on $\varepsilon$ and $\gamma$ because it is given by the integral of training dynamics. It seems hard to obtain a concrete value of this integral. Instead of evaluating each entries of $c$, let us analyze the average value of $c$, that is, $\|c\|_1/d = \sum_{i=1}^d c_i/d$. This approach gives us some insight into a typical value of the exponent $c$:

$$\|c\|_1/d = \frac{1}{d}\int_0^\infty r(s)^\top(XX^\top)r(s)ds \tag{S.47}$$

$$\ge \frac{4n}{d}\lambda_{min}(XX^\top)\underline{R}. \tag{S.48}$$

A similar evaluation has been used in the analysis of SAM (Andriushchenko & Flammarion, 2022).

### A.4 HESSIAN

The MSE loss of the diagonal linear network has the following Hessian:

$$H = \frac{1}{n}\left(\text{diag}(\tilde{X}^\top r) + 2\text{diag}(w)\tilde{X}^\top\tilde{X}\text{diag}(w)\right). \tag{S.49}$$

At the interpolation solution,

$$H = \frac{2}{n}\text{diag}(w)\tilde{X}^\top\tilde{X}\text{diag}(w). \tag{S.50}$$

## B DERIVATION OF THEOREM 5.1

It is straightforward to derive this theorem. Consider the time step $t$ satisfying $L(\theta_t) < b$ and $L(\theta_{t+1}) > b$. The update rule is given by

$$\theta_{t+1} = \theta_t + \eta\nabla_\theta L(\theta_t), \tag{S.51}$$

$$\theta_{t+2} = \theta_{t+1} - \eta\nabla_\theta L(\theta_{t+1}). \tag{S.52}$$

Taking the summation, we get

$$\theta_{t+2} = \theta_t - \eta(\nabla_\theta L(\theta_{t+1}) - \nabla_\theta L(\theta_t)) \tag{S.53}$$

$$= \theta_t - \eta^2 \frac{\nabla\mathcal{L}(\theta_t + \eta\nabla\mathcal{L}(\theta_t)) - \nabla\mathcal{L}(\theta_t)}{\eta}. \tag{S.54}$$

Similarly, for $L(\theta_t) > b$ and $L(\theta_{t+1}) < b$, we have

$$\theta_{t+1} = \theta_t - \eta\nabla_\theta L(\theta_t), \tag{S.55}$$

$$\theta_{t+2} = \theta_{t+1} + \eta\nabla_\theta L(\theta_{t+1}). \tag{S.56}$$

and get

$$\theta_{t+2} = \theta_t + \eta(\nabla_\theta L(\theta_{t+1}) - \nabla_\theta L(\theta_t)) \tag{S.57}$$

$$= \theta_t - \eta^2 \frac{\nabla\mathcal{L}(\theta_t) - \nabla\mathcal{L}(\theta_t - \eta\nabla\mathcal{L}(\theta_t))}{\eta}. \tag{S.58}$$

## C EXPERIMENTS

### C.1 COMPUTATION OF LEARNING WITH GR

#### C.1.1 PSEUDO-CODE AND IMPLEMENTATION

In the experiments on benchmark datasets, we computed the GR term in each mini-batch of SGD update. The pseudo-code for F-GR is given in Algorithm 1. The double backward computation is implemented as shown in Listing 1.

---

**Algorithm 1** Learning with F-GR

---

**Input:** mini-batches$\{B_1, ..., B_K\}$
1: **while** SGD update **do**
2:     **if** $i$-th mini-batch **then**
3:         $\Delta\mathcal{L} \leftarrow \nabla\mathcal{L}(\theta; B_i)$
4:         $\theta' \leftarrow \theta + \varepsilon\Delta\mathcal{L}$
5:         $\Delta\mathcal{L}' \leftarrow \nabla\mathcal{L}(\theta'; B_i)$
6:         $\Delta R \leftarrow (\Delta\mathcal{L}' - \Delta\mathcal{L})/\varepsilon$
7:         $\theta \leftarrow \theta - \eta(\Delta\mathcal{L} + \gamma\Delta R)$
8:     **end if**
9: **end while**

---

```
1  ...
2  loss.backward(create_graph=True) #backpropagation of original loss
3  loss_DB = (gamma/2)*sum([torch.sum(p.grad**2) for p in model.parameters()
     ]) #computing GR term
4  loss_DB.backward() #backpropagation of GR term
5  optimizer.step()
6  ...
```

Listing 1: Implementation of DB in PyTorch.

#### C.1.2 EVALUATION ON THE NUMBER OF MATRIX MULTIPLICATION

We represent an $L$-layer fully connected neural network with a linear output layer by $A_l = \phi(U_l)$, $U_l = W_l A_{l-1}$ for $l = 1, ..., L$. We define the element-wise activation function by $\phi(\cdot)$ and weight matrix by $W_l$. For simplicity, we neglect bias terms. Note that we have multiple samples $A_0$ (within each minibatch) as an input and $W_l A_l$ requires a matrix-matrix product. Therefore, the forward pass requires $L$ matrix multiplication. Next, let us overview usual backpropagation on the forward pass $\{A_0 \to A_1 \to \cdots \to A_L\}$. We can express the backward pass as $B_l = \phi'(U_l) \circ (W_{l+1}^\top B_{l+1})$, where the backward signal $B_l$ corresponds to $\partial\mathcal{L}/\partial U_l$ ($l = 1, ..., L-1$). Then, the backward pass requires

$L-1$ matrix-matrix multiplication between weights $W$ and backward signals $B$. In addition, we need to compute the gradient $\partial \mathcal{L} / \partial W_l = B_l A_{l-1}^\top$ for $\nabla \mathcal{L}$ and this is also a matrix-matrix multiplication. Alter all, we need $3L-1$ matrix multiplication for $\nabla \mathcal{L}$.

**Finite difference computation:** $\nabla \mathcal{L}(\theta')$ requires the same number of matrix multiplication as the normal backpropagation. Therefore, $\nabla \tilde{\mathcal{L}}$ requires $6L-2$. For a sufficiently deep network, this is $\sim 6L$.

**Double Backward computation:** Let us denote $\partial \mathcal{L} / \partial W_l$ by $G_l$. Figure 2 represents the forward pass for computing the gradient of GR. Note that the upper part of this graph, i.e., $\{A_0 \rightarrow A_1 \rightarrow \cdots \rightarrow B_L \rightarrow \cdots \rightarrow B_1\}$, is well-known in double backpropagation of $\nabla B_1$ for the input-Jacobian regularization. As explained in Drucker & Le Cun (1992), the computation of $\nabla B_1$ is equivalent to apply backpropagation to this upper part of the graph. GR requires additional $L$ nodes for $G_l$. Note that when we have a forward pass with matrix multiplication, its backward computation requires two matrix multiplications. That is, when a node of the forward pass $S$ is a function of the matrix $X$ given by $X = UV$, we need to compute $\partial S / \partial U = (\partial S / \partial X)V$ and $\partial S / \partial V = U(\partial S / \partial X)$ in the backpropagation. In addition, we do not need to compute the derivative of $A_0$. After all, we need $2 \times (3L-1) - 2 = 6L-4$ for the $\nabla R$. Since we also compute the gradient of the original loss $\nabla \mathcal{L}$, we need $9L-5$. For a sufficiently deep network, this is $\sim 9L$.

## C.2 DETAILS OF EXPERIMENTAL SETTINGS

**Figure 1:** We trained MLP (width 512) and ResNet on CIFAR-10 by using SGD with GR. We set batch size 256, momentum 0.9, initial learning rate 0.01 and used a step decay of the learning rate (scaled by 5 at epochs 60, 120, 160), $\gamma = \varepsilon = 0.05$ for GR. We showed the average and standard deviation over 5 trials of different random initialization.

**Figure 3:** (a) We trained the 4-layer MLP and ResNet-18 on CIFAR-10 by using SGD with GR. We trained the models with various hyper-parameters $\varepsilon = \{10^{-5}, 5 \times 10^{-5}, ..., 0.5, 1\}$ and $\gamma = \{10^{-4}, 2 \times 10^{-4}, 5 \times 10^{-4}, 10^{-3}, ..., 1, 2, 5\}$. The other settings are the same as in Figure 1. We set batch size 128, weight decay 0.0001, and used no other regularization technique or data augmentation.

**Figure 4:** We generated synthetic data by $x_i \sim \mathcal{N}(\mu 1, \sigma^2 I)$ and $y_i \sim \mathcal{N}(\langle \beta^*, x_i \rangle, 0.01)$. $\beta^*$ is $k^*$-sparse with non-zero entries equal to $1/\sqrt{k^*}$. We set $d = 100$, $n = 50$, $\mu = \sigma^2 = 5$, $\gamma = 0.02$ and initialization $\alpha_{0,i} \sim \mathcal{N}(0, 0.01)$. We trained the models by the discrete time update with a small learning rate $\eta = 0.001$. We showed the average of 25 trials with different seeds. We trained the models until the training loss $\mathcal{L}$ became lower than $10^{-8}$.

**Figure 5:** We trained ResNet-18 on CIFAR-10 by SGD with flooding ($b = 0.05$). The setting is the same as in Figure 1. We computed the gradient norm $R$ by the average of mini-batches in each epoch. We showed the average and standard deviation over 10 trials of different random initialization.

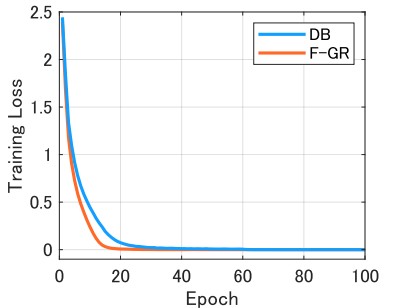 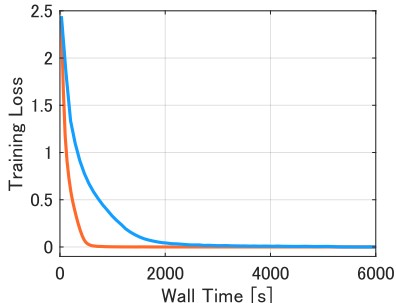

Figure S.1 : Training dynamics in ResNet-18 on CIFAR-10. Learning with F-GR is much faster in wall time.

## C.3 ADDITIONAL EXPERIMENTS

### C.3.1 TRAINING DYNAMICS

**Figure S.1:** This figure shows the trajectories of the original training loss $\mathcal{L}$ during the training. Its setting is the same as in Figure 1. We observed that learning with F-GR could make the loss decrease faster than DB in the sense of convergence rate (i.e., the number of epochs). This means that the loss converges even faster in wall time.

### C.3.2 GENERALIZATION PERFORMANCE

**Figure S.2:** To see the difference among algorithms in more detail, we show test accuracy along $\varepsilon$ axis with a fixed $\gamma$ of the grid search shown in Figure 3. Each line represents the average and standard deviation over 5 trials of different random initialization. We fixed $\gamma = 0.5$ for MLP and $\gamma = 0.05$ for ResNet-18. This means that the objective function is the same among different algorithms. Nevertheless, the eventual performance is different. For a large $\varepsilon$, F-GR achieves the higher test accuracy than DB beyond one standard deviation. For such a large $\varepsilon$, F-GR also performs better than B-GR.

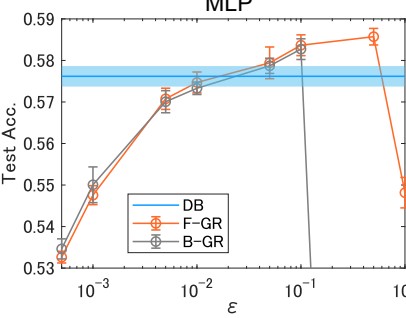 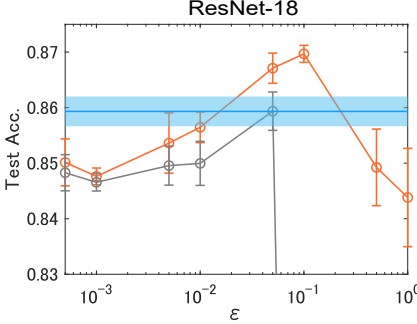

Figure S.2 : Test accuracy shown in Figure 3 along $\varepsilon$ axis with a fixed $\gamma$. We fixed $\gamma = 0.5$ for MLP and $\gamma = 0.05$ for ResNet-18.

**Figure S.3:** This figure shows an additional experiment of the grid search shown in Section 3.3. We did experiments on a different architecture and dataset, that is, ResNet-34 on CIFAR-10. The result is consistent with those in Figure 3. Learning with F-GR achieves the highest accuracy for large ascent steps. B-GR performs much worse for them. In addition, the highest accuracy of F-GR is better than that of DB. The best test accuracy was $(\text{F-GR}, \text{B-GR}, \text{DB}) = (59.9, 58.6, 59.5) \pm (0.5, 0.4, 0.5)$. From this experiment, we can see that the result of the finite difference computation with small $\varepsilon$ does not necessarily coincide with that of DB. We observed that the training dynamics showed instability for too small $\varepsilon$. This would be attributed to numerical instability. The important point is that F-GR shows better accuracy than DB for large ascent steps.

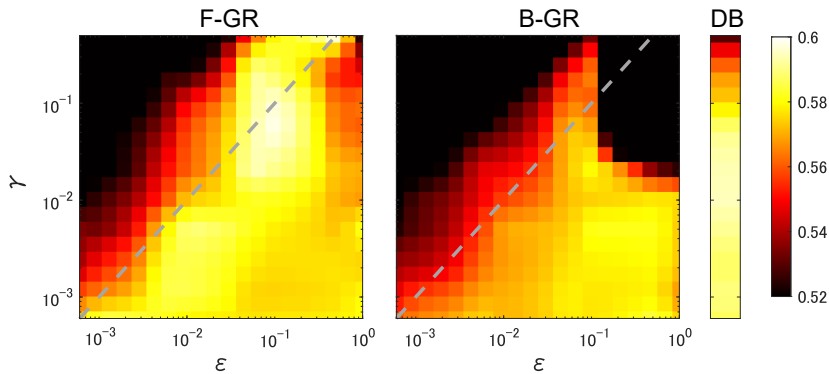

Figure S.3 : Grid search on learning with different GR algorithms in ResNet-34 on CIFAR-100. The color bar shows the average test accuracy over 5 trials. Gray dashed lines indicate $\gamma = \varepsilon$.

**Table S.1**: We trained WideResNet-28-10 (WRN-28-10) with $\gamma = \{0, 10^{-4}, 10^{-3}, 10^{-2}, 10^{-1}\}$. For F-GR and B-GR, we set $\epsilon = \{0.001, 0.01, 0.1\}$. We computed the average and standard deviation over 5 trials of different random initialization, and reported the best average accuracy achieved over all the above combinations of hyper-parameters. F-GR performs better than DB and B-GR beyond one standard deviation in most cases. We used crop and horizontal flip as data augmentation, cosine scheduling with an initial learning rate 0.1, and set momentum 0.9, batch size 128, and weight decay 0.0001.

Table S.1 : Test accuracy of WRN-28-10 shows that F-GR performs better. We trained the models with/without data augmentation (DA).

| | WRN-28-10 | | | |
| | CIFAR-10 | | CIFAR-100 | |
| | w/o DA | w/ DA | w/o DA | w/ DA |
|---|---|---|---|---|
| F-GR | $92.1_{\pm 0.2}$ | $96.1_{\pm 0.1}$ | $71.3_{\pm 0.3}$ | $80.7_{\pm 0.2}$ |
| B-GR | $91.9_{\pm 0.1}$ | $95.9_{\pm 0.1}$ | $71.1_{\pm 0.5}$ | $80.2_{\pm 0.2}$ |
| DB | $91.7_{\pm 0.2}$ | $95.9_{\pm 0.1}$ | $70.3_{\pm 0.2}$ | $80.3_{\pm 0.4}$ |

### C.3.3 DIAGONAL LINEAR NETWORK

**Figure S.4**: We trained DLNs with various $\varepsilon$ and $\gamma$ in the same setting as in Figure 4. The black circles in the figure show the cases of the lowest test error. The best test error was $(\text{F-GR}, \text{B-GR}) = (10^{-1.37}, 10^{-1.16})$ and F-GR performed better.

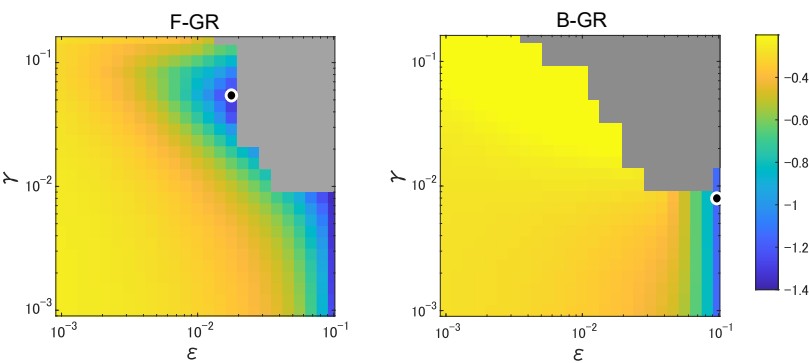

Figure S.4 : Learning of diagonal linear networks with GR. The color bar shows the average test loss over 10 trials. Training dynamics exploded in the gray area.

### C.3.4 FLOODING METHOD

**Figure S.5**: This figure confirms at which epoch the training loss started to get close to the flood level. The experimental setting is the same as in Figure 5. The blue line shows a flip rate, that is, the ratio of how many times the training loss gets smaller than the flood level during each epoch. Around the 10-th epoch, the training loss started to reach the flooding level and the gradient norm also started to decrease.

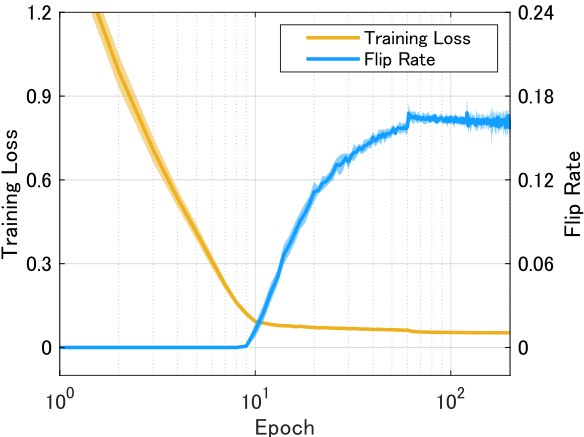

Figure S.5 : Flip rate of flooding with $b = 0.05$.

