# OpenReview forum: "Understanding Gradient Regularization in Deep Learning: Efficient Finite-Difference Computation and Implicit Bias"
_ICLR.cc/2023/Conference — Submitted to ICLR 2023_

### Official Review · Reviewer_hFFP · 2022-10-24

**Confidence:** 3
**Correctness:** 2
**Technical Novelty And Significance:** 3
**Empirical Novelty And Significance:** 3
**Recommendation:** 5

**Clarity, Quality, Novelty And Reproducibility:**

Clarity is good except for the theory part.

Quality could have been improved by providing more details on the experimental results like std.

Novelty is very good. In fact it is too good and requires a careful examination in my prespective.

Reproducibility is fine.

**Strength And Weaknesses:**

# Strength
1. Overall I find the writing is good and most of the presentation is clear.
2. It is very novel to my knowledge the observation/conjecture that a finite difference scheme with a positive and large time step might bring additional implicit bias.
3. I like the discussion about the connection of the authors' finding and the existing SAM, flooding methods. I think it brings some interesting insights that could potentially unify several existing methods that were developed from different areas.

# Weakness

The Strength#2 is too surprising to me such that I have to ask a lot of questions to be fully convinced.
1. Section 3.3. Could you report the stand deviation/error bar for the data obtained for MLP and ResNet-18?

2. I checked Table S.1 that presents test accuracy for WRN, where some error bar is provided. However, the difference between the three schemes is too small to be interesting to be honest (most of the cases the differences is O(0.1%) in terms of accuracy).

3. Figure 3 is interesting, but is hard to interpret as the exact numbers are hard to read.

4. Note that compared to DB, finite difference introduces an additional hyper parameter $\\epsilon$. Would you say that the small advantage of finite difference is actually due to the additional flexibility for tuning the new hyperparameter? I would love to hear some discussion on this issue, because the improvement does not seem to be significant so I think it is very likely caused by tuning the hyperparameter.

The following questions are about the theory, which I find not been presented as clear as the other part of the paper.

5. Theorem 4.1. Assumption (iiI) could use more discussions. First of all, $L(\\cdot)$ is a function of a $2d$-dimensional vector, why it can be integral over a real line? Secondly, it requires the upper and lower bound for the integral to be independent of small $\\gamma$ and $\\epsilon$. Why this is related to the convergence speed of the dynamics with/without GR?

6. Can you briefly discuss how large the quanatity $\\epsilon$ and $\\gamma$ could be in Theorem 4.1? Because in the follow-up discussion you have mentioned that a larger $\\epsilon$ leads to a stronger regularization. I am just wondering how much changing on the bound is allowed to happen.

7. Since Theorem 4.1 only provides upper bound on $\\alpha$, this does not necessarily support the follow-up discussion that a larger $\\epsilon$ leads to a stronger regularization. Note that a larger $\\epsilon$ only leads to a smaller upper bound on $\alpha$, which does not necessarily mean that $\alpha$ itself is decreased. Indeed, Eq(12) provides another upper bound on $\\alpha$ that does not decrease when $\epsilon$ increases. So at least one can take the minimum of the bounds in Eqs(11) and (12), which is still a valid upper bound on $\alpha$. Would you say that this improved upper bound is still decreasing when $\\epsilon$ increases?

8. It seems that $c$ in Eq(12) is a vector? But $c\^\*$ in Eq(11) is a constant? I am just confused by the notations, and I find the follow-up discussion after Proposition 4.2 is hard to interpret because it mentioned "the average over entries" of $c$.


# Typos
- A lot of figures are not correctly numbered. Please double check.
- Theorem 4.1, assumption (ii), upper -> upper bound.
- The comments on Figure 5 might have some typos. It seems to me that the loss gets close to the flood level at about the $100$ epoch instead of the $10$-th epoch.







**Summary Of The Paper:**

The work considers a gradient norm regularizer and the implicit bias afforded by the numerical methods for computing the regularization terms, namely, (1) forward finite difference, (2) backward finite difference, or (3) double back propagation. It was suggested that forward finite difference with a relatively large time step gives the best generalization among the three methods. Additionally, the paper discussed some theory for this phenomenon using a diagonal linear networks. Finally, some connections to two existing algorithms, sharpness-aware-minimization and flooding method, are discussed.

**Summary Of The Review:**

Please see above. Due to the mentioned issues I would like to keep my scoring conservative for now. I will consider increasing my score if my questions receive proper replies. I am being conservative because some of the results are quite surprising to me.

---

> ### Author Response · Authors · 2022-11-19
> **Response to Reviewer hFFP**
>
> Thanks for your insightful and constructive feedback. Before giving our detailed replies, we would like to emphasize that our main purpose is not to achieve a large improvement in test accuracy but to clarify the difference between algorithms.
>
> > 1. Section 3. 3… stand deviation/error bar
>
> We have added the best of average test accuracy and its standard deviation to the revised manuscript. It is $(\text{F-GR},\text{B-GR},\text{DB})=(58.6,58.3,57.6)\pm(0.2,0.2,0.2)$ for MLP and $(87.0,86.2,86.3)\pm(0.2,0.3,0.3)$ for ResNet-18. Note that before the revision, in this section, we reported the best accuracy  (not average) among trials. As you pointed out, it would be better to show the standard deviation and we decided to replace it with the average one.
>
> >  2. Table S.1 … the difference between the three schemes is too small to be interesting
>
> After reading your comment, we realized that the standard deviation was relatively large (e.g. $69.7 \pm 0.7$). This was because we reported the test accuracy at the last epoch in this table. Since we reported the best accuracy during the whole training in other figures, we have replaced Table S.1 with the latter one. In the new Table S.1, we can see that the standard deviation got smaller in some cases, especially, in the cases of w/o DA. F-GR got higher average accuracy than B-GR and DB, and in some cases, beyond one standard deviation. Although it is still in the range of $O(0.1)$, the superiority of F-GR on the best average accuracy is consistent in all cases.
>
> > 3. Figure 3 is interesting, but is hard to interpret
>
> We have added a new Figure S.2 which clarifies the change of the accuracy along the $\varepsilon$ axis with a fixed $\gamma$. Despite the same objective function (i.e., with the same $\gamma$), each algorithm shows different behaviors as we expected.
>
> > 4. advantage of finite difference due to the additional flexibility
>
> The finite difference shows some consistent tendencies for the change of $\varepsilon$. For example, we can see a consistent peak structure of test accuracy for F-GR in Figures 3 and S.3. In addition,
> Figure S.2 confirms that F-GR achieves higher accuracy than DB beyond more than two standard deviation ranges for a large $\varepsilon$. We guess that your additional flexibility means an increase in the number of trials, or “rolling dice more”. The probability that the above consistent tendencies hold would be very low if they were totally caused by the additional flexibility.
>
> > 5. Theorem 4.1. Assumption (iii)
>
> Our insufficient notation/explanation caused the confusion. First, Assumption (iii) means the integral over time. We fixed it. Second, what we meant to say is that
> “the upper bound of Assumption (iii) requires that the convergence speed does not get too small for sufficiently small $\varepsilon$ and $\gamma$“. If it becomes too small, the integral may diverge (e.g.,  $\mathcal{L}(w(t)) \sim 1/t$). We fixed the misleading sentence.
>
> >6. how large the quantity $\varepsilon$ and $\gamma$ could be
>
> We added new inequalities (S.29) and (S.36) and some discussion for them in the revised manuscript. Ineq. (S.29) ((S.36), respectively) implies that we need a smaller $\gamma$ ($\varepsilon$) for a larger norm of parameters or a slower convergence of dynamics. It will be interesting to further explore such non-asymptotic bounds, but note that the main theme of the current work is the asymptotic behavior of the first-order term.
>
> >7. only provides upper bound on $\alpha$
>
> Certainly, the upper bound does not necessarily support the monotonic decreasing of $\alpha$ (e.g. $\tanh(\varepsilon ) \exp(-\varepsilon ) \leq \exp(-\varepsilon )$). In more detail, the whole upper bound comes from (S.14), where $\Psi_0$ corresponds to $c$ in (12) and $\Psi_1$ to $c^*$ in (11). This means that the whole upper bound is given by a product, that is, $\alpha_{F-GR} \leq \alpha_0 \circ f(\varepsilon) \circ \exp(-\gamma \varepsilon c^*) $ where $ f(\varepsilon)$ may increase for $\varepsilon$.
> However, since we have $ f(\varepsilon)  \leq 1$ for any $\varepsilon$, $\exp(-\gamma \varepsilon c^*) $ works as the upper bound. Thus, as long as we focus on the first-order term and neglect $O(\varepsilon^2)$ and  $O(\gamma^2)$, the upper bound decreases as the $\varepsilon$ increases.
> Let us also remark that we empirically confirmed the monotonic increase of the implicit bias to the rich regime in Figures 4 and S.4.
> Although the upper bound is an indirect evaluation of $\alpha$, we believe that it would be quite interesting to identify the theoretical subject consistent with experiments.
>
> >8. $c$ in Eq(12) is a vector?
>
> Both $c$ and $c^*$ are d-dimensional vectors. We emphasize more the dimensionality in the revised manuscript.
>
> >  gets close to the flood level at about the 100 epoch
>
> Because we used a step decay of the learning rate, we can see a small decrease around 100-th. Our new figure S.5 clarifies that the flooding starts around the 10-th epoch.

---

### Official Review · Reviewer_jCXs · 2022-10-25

**Confidence:** 4
**Correctness:** 4
**Technical Novelty And Significance:** 3
**Empirical Novelty And Significance:** 3
**Recommendation:** 6

**Clarity, Quality, Novelty And Reproducibility:**

How can we better understand Theorem 4.1 as GR has an implicit bias to select the L1 solution? From the closed-form expression, we find an exponential shrinkage of $\alpha$. While for L1 solution, the coefficient should be sparse if I am correct.

It is interesting to see why forward finite-difference method performs better than the backward one. Meanwhile, different discretization approximation methods have very distinct performance dependence on learning rate; it would be helpful to provide any intuition behind these observations. Does the theory in Section 4 justifies the observation in Section 3.3?

Apart from aforementioned questions and concerns, the paper is easy to follow, and the results are correct.

**Strength And Weaknesses:**

==================== Strength ====================

The paper targets at an important problem of circumventing expensive Hessian computations in regularized training. Regularized training is essential to the success of modern learning models and gradient regularization has close ties to the flatness of minima and generalization properties. The finite-difference method resolves the computational issue and also achieves good -- even better -- performance. This should be of high interest to practitioners.

The paper is well motivated and clearly written. The theoretical results seem to be correct.

The scope of the paper is actually broad; it aims to provide a comprehensive understanding of GR, touching its implementation, implicit bias, and connection to other popular methods.

==================== Weakness ====================

Section 4 is rather toy and seems to lack technical contributions (compared to Woodworth et al. (2020)).

**Summary Of The Paper:**

GR is a method that penalizes the gradient norm of the training loss during training. However, computing the gradient of GR objectives leads to Hessian evaluation, which is computationally expensive. The paper attempts to accelerate the gradient regularized (GR) training by finite-difference approximations. In particular, forward and backward finite-difference methods are discussed. Theoretically, the implicit bias of GR in diagonal linear model is explicitly characterized. The paper also connects GR to other algorithms based on iterative ascent and descent steps for exploring flat minima.

**Summary Of The Review:**

I am leaning towards a positive rating, due to the clarity and the broad scope of the paper.

A major criticism might be Section 4 is a bit disconnected, as the implicit bias does not provide deeper understanding of the finite-difference implementation of GR. This is partly due to the limited analytical tools. Therefore, I would recognize Section 4 as some understanding of GR. In this sense, the concept in the paper is broad, while the inner coherence can still be improved.

==================== Post author response ====================

Thank you for addressing my concerns and I am willing to keep the initial rating. A minor point is that I am still not very clear what are the technical contributions in theory. The analysis handles the new term $\Psi_1$, but what are the new proof details or tools leveraged (developed) in paper? Providing these details would be even more helpful.

---

> ### Author Response · Authors · 2022-11-19
> **Response to Reviewer jCXs**
>
> Thanks for your insightful feedbacks and acknowledging the interesting points of our work.
>
> >Section 4 is rather toy and seems to lack technical contributions (compared to Woodworth et al. (2020)).
>
> We would like to emphasize that our Lemma A.1 enabled us to analyze a completely new term $\Psi_1$ (appeared in (S.14)) compared to the previous works. In more detail, compared to the original work of Woodworth et al. (2020), Andriushchenko & Flammarion (2022) analyzed $\Psi_0$, and ours analyzed $\Psi_1$. As the techniques developed for evaluating higher-order terms (e.g., $\Psi_0$, $\Psi_1$), it broadened the range of learning algorithms that we could analyze in DLNs. We believe that this progress in analytical techniques will further enrich the usefulness of DLNs as a solvable model.
>
> >How can we better understand Theorem 4.1 as GR has an implicit bias to select the L1 solution? … the coefficient should be sparse
>
> Concerns about this question seem not unique to our study of GR but common to the previous studies of learning algorithms in DLNs. Previous work has revealed that $\alpha$ decreases by a larger learning rate in the discrete update [Nacson et al. 2022], the noise of SGD [Pesme et al. 2021], and SAM update [Andriushchenko & Flammarion, 2022]. These studies focus on the change of $\alpha$ caused by the algorithms and the evaluation of the sparseness is not explicit as is similar to our study. For example, to explicitly characterize the sparseness of the coefficient $\beta$, we need to count the number of zero entries (i.e., L0 norm). But, such evaluation seems an open problem in DLNs. After all, what we can understand in the current situation is the shape of the implicit regularization term, that is, a smaller (larger, respectively) $\alpha$ changes the shape of $\phi_\alpha (\beta)$, that is, the ridge-less regularizer, and makes it close to L1 (L2) norm.
>
> >It is interesting to see why the forward finite-difference method performs better than the backward one … theory in Section 4 justifies the observation in Section 3.3?
>
> Our results suggest two potential reasons. First, as is suggested by the theory in Section 4, forward and backward methods can have different implicit biases. As the previous studies on DLNs supposed,  the rich regime corresponds to feature learning and we can expect the improvement of generalization. Actually, we can see that F-GR achieves higher generalization performance than B-GR in the training of ResNet-18 (for example, see Figure S.2 right). We can also see that in broad areas of the grid search (Figures 3, S.3), F-GR performs better than B-GR. Thus, the theory in Section 4 seems to justify the observation in Section 3.3.
>
> Second, as is shown in the grid search of hyper-parameters (Fig. 3, S.3 in Section 3.3), the performance of B-GR rapidly becomes worse for a large $\varepsilon$. This is because the training dynamics of B-GR is more likely to get unstable than F-GR. We can also observe the same tendency for the experiments on DLNs (Figure S.4). But unfortunately, the theoretical analysis in Section 4 is based on the perturbation and assumes a small $\varepsilon$. Therefore, it is non-trivial that the theory justifies the whole difference between F-GR and B-GR, but at least, empirically, the tendency for a large $\varepsilon$ is also common between experiments on realistic networks and DLNs.

---

### Official Review · Reviewer_MSVy · 2022-10-25

**Confidence:** 3
**Correctness:** 4
**Technical Novelty And Significance:** 3
**Empirical Novelty And Significance:** 3
**Recommendation:** 6

**Clarity, Quality, Novelty And Reproducibility:**

This paper has good clarity, quality, and novelty. The theoretical analysis is new and is useful for understanding the role of gradient regularization.


**Strength And Weaknesses:**

Strength:
1. This paper demonstrates certain advantages of using finite-difference computation over double backpropagation on the gradient regularization term.
2. This paper provides a theoretical analysis to study the implicit bias of gradient regularization on a diagonal linear network and shows that GR can potentially lead to better solutions.
3. This paper further builds a connection between GR and other learning methods.

weakness:
1. The theoretical analysis is not clear to me. In particular, Theorem 4.1 and  Proposition 4.2 only give bounds on the final solutions found by the algorithms, while their sharpness is not fully justified. So I am still not clear what’s the exact quantity of the solution.
2. Following the previous comment, why the final solutions found by F-GR will be smaller than $\alpha_0$? Does it require $\alpha_0$ to be positive?
3. I also cannot understand why Theorem 4.1 suggests that GR tends to select the L1 solution and why the L1 solution has a better generalization performance. It may need a rigorous theory to connect the implicit bias of GR and the generalization performance.
4. When performing experiments on CIFAR, do you add standard weight decay regularization? More details of the experiment setup should be added.


**Summary Of The Paper:**

This paper studies the algorithmic behavior of gradient regularization in deep learning. In particular, this paper considers three versions of practical implementations of the gradient calculation of the gradient regularization term: forward finite-difference, backward finite-difference, and double backpropagation. The authors theoretically investigate the algorithmic bias of these three implementations on a diagonal linear network and prove that gradient regularization has an implicit bias to find the solution with potentially better generalization. Finally, the authors illustrate the close connection between finite-difference GR and other training methods such as SAM and the flooding method.

**Summary Of The Review:**

Overall this is a good paper. However, both the theoretical and experimental parts need to be improved according to my comments in the weakness section.

---

> ### Author Response · Authors · 2022-11-19
> **Response to Reviewer MSVy**
>
> We would like to thank the reviewer for the helpful feedbacks, and for acknowledging the interesting points of our work.
>
> >1. still not clear what’s the exact quantity of the solution.
>
> We would have two points to clarify your concerns. First, note that the studies on implicit bias in DLNs mainly consider only the change of $\alpha$ caused by the change of learning algorithms.  That is, the previous works in learning algorithms claimed a shrinkage of $\alpha$ as the implicit bias to the rich regime (i.e., sparse solution caused by near L1 regularization), for example, a larger learning rate in the discrete update [Nacson et al. 2022], noise of SGD [Pesme et al. 2021], and SAM update [Andriushchenko & Flammarion, 2022]. Our theorem follows a line of this evaluation.  Instead of explicitly evaluating the eventual value of the solution $\beta$, we evaluate the shape of $\phi_\alpha (\beta)$ (i.e., the implicit ridge-less regularizer).
>
> The second point is unique to our study, that is, the upper bound evaluation. To clarify the meaning of the bound, we think that answering your second question seems essential.
>
> >2. Following the previous comment, why the final solutions found by F-GR will be smaller than α0?
>
> For usual gradient descent without GR, $\alpha$ is equal to $\alpha_0$. By adding the explicit GR, the training dynamics changes, and its contribution is reduced to the size of $\alpha$. In particular, F-GR makes $\alpha$ smaller than $\alpha_0$ and this means that F-GR contributes to making the solution biased to the kernel regime compared to training without F-GR if we start the training from the same initialization scale $\alpha_0$. In other words, to obtain the same strength of the implicit regularization in the training without GR, we need to set a smaller initialization scale $\alpha_0$.
>
> >Does it require α0 to be positive?
>
> Positive $\alpha_0$ is a common assumption used in the studies of DLNs (e.g. Theorem 1 in [Woodworth et al. 2020], [Andriushchenko & Flammarion 2022]). This is for avoiding a complicated notation and it is easy to consider a general case. Note that $\alpha$ appears as $\alpha^2$ in the main formulation of the implicit bias (9). Therefore, the sign of $\alpha$ is not essential and its absolute value plays a fundamental role. For example, it is rather straightforward to see $|\alpha_{F-GR}|  \lesssim |\alpha_0| \circ \exp(- \gamma \varepsilon c^*)$.
>
> >3.  why Theorem 4.1 suggests that GR tends to select the L1 solution
>
> As is similar to our reply to your first question, it would be helpful to remark on the current situation of the studies of DLNs. In general, a smaller $\alpha$ makes the shape of the function $\phi_\alpha(\beta)$, that is, the ridge-less regularizer, close to L1 norm (i.e., $\|\| \beta \|\|_1$). It may be also informative to check the shape of $\phi_\alpha(\beta)$ visualized in Figure 2 of [Woodworth et al. 2020]. It can be regarded to as an interpolation between L1 and L2 norm.
> The exact explicit evaluation of the final solution $\|\| \beta \|\|_1$ seems still limited (as a side note, an indirect evaluation is given in Theorem 2 of [Woodworth et al. 2020] in special cases).
> Therefore, the studies on the implicit bias caused by learning algorithms mainly focus on the shrinkage of $\alpha$.
>
>
> > why the L1 solution has a better generalization performance
>
> This is also an assumption or hypothesis that the previous works on DLNs supposed. In a realistic dataset like natural images, sparseness is known as one candidate to explain well the statistical properties of data. Feature learning (that is outside of kernel regime) is believed to work well in such a data structure. Therefore, all previous works and ours did experiments on DLNs by using data generated as a sparse regression problem (i.e., $y\sim \langle \beta^*,x \rangle$ where $\beta^*$ is sparse). We can expect that the rich regime achieves better generalization because it matches this data structure. Theoretical evaluation on the generalization in DLNs will be an interesting research direction but still limited (for example, only empirical evidence is shown in [Woodworth et al. 2020], [Andriushchenko & Flammarion 2022]).
>
> >4. When performing experiments on CIFAR, do you add standard weight decay regularization? More details of the experiment setup should be added.
>
> We have summarized the detailed experiment setup in Sections C.2 and C 3. The weight decay was added in the experiments in Figure 3 (MLP and ResNet-18 in CIFAR-10) and Table S.1 (WRN). Its strength was set to 0.0001. Adding weight decay or not makes no qualitative difference, though.

---

### Official Review · Reviewer_V6uG · 2022-10-26

**Confidence:** 3
**Correctness:** 4
**Technical Novelty And Significance:** 2
**Empirical Novelty And Significance:** 2
**Recommendation:** 5

**Clarity, Quality, Novelty And Reproducibility:**

I did not understand the following aspect: why is $\frac{\gamma}{2}R(\theta)$ used in equation (1) instead of $\frac{\gamma}{4}R(\theta)$? If I understood correctly, the latter was shown to correspond to the implicit bias of SGD in equation (20) of (Smith et al., 2021) when $\gamma$ is the stepsize.


**Strength And Weaknesses:**

## Strengths
1. Gradient regularization has been popular recently and the paper might be of interest to the conference attendees.
2. The finite-difference approximation is indeed cheaper to calculate than the full gradient with backpropagation.
3. Some of the theoretical claims are checked numerically.

## Weaknesses
1. The overall intuition behind the efficiency of finite-difference approximation is not clear. In general, finite differences yield very noisy estimates of gradients that poorly scale with dimension, and it is not explained why in this case this approximation happens to be better. As far as I can see, this issue, which is normally the main counter-argument against finite-differences, is not addressed in the paper.
2. The reduction of double backpropagation to forward passes is not a critical improvement. It is of course nice to reduce the training time, but the impact of this change is limited and is only worth it if all other factors (such as variance) remain the same.
3. The diagonal linear network is mostly a toy example that no one uses in practice. On top of that, the derivation for this problem is mostly the same as done in prior works. A more general theory, for instance, for locally smooth objectives, could be of more interest.

**Summary Of The Paper:**

Update: I raised my score to 5 after reading the authors' rebuttal.

--------
The paper studies gradient regularization, a technique that is often associated with improved generalization in deep learning. The authors consider a finite-difference approximation of gradient regularization that not only decreases the computational cost but also is shown to improve generalization empirically. This effect is proven to hold for diagonal linear networks, which can be seen as a very simple example of neural networks. The authors also validate their claims using numerical experiments on neural networks.

**Summary Of The Review:**

All in all, this paper presents a number of small results about finite-difference schemes for gradient regularization. Each result on its own seems to be quite small. The results do not become much stronger together either, since they are on orthogonal topics: numerical efficiency, implicit bias on diagonal linear networks, and the relation to other deep learning techniques. The work seems to lack depth in its theoretical investigation, which makes me suggest a rejection.

---

> ### Author Response · Authors · 2022-11-19
> **Response to Reviewer V6uG**
>
> We would like to thank the reviewer for the useful comments and suggestions. First, let us reply to your comments on our theoretical results:
>
> >3. The diagonal linear network is mostly a toy example that no one uses in practice.
>
> The previous works have established the significance of the diagonal linear network (DLN) and it is common to use it as a solvable model.  As we cited, the studies on DLN contributed to giving theoretical insight into the empirical success of algorithms, for example, discrete update (Nacson et al., ICML 2022) and SAM (Andriushchenko et al., ICML 2022).
> Certainly, if we have more general frameworks to analyze these algorithms, focusing only on DLN might be a limited contribution.  However, we have no theory for GR yet, the results on DLNs are valuable and expected to serve as a starting point for further research.
> Therefore,  using DLN cannot become a flaw in our paper.
>
> >On top of that, the derivation for this problem is mostly the same as done in prior works.
>
> Actually, it is not the same. As we mentioned in Section 4.2,  “learning with F-GR requires evaluating a novel c* term, which has not appeared in the analyses of previous studies.” In more detail, one previous work investigated $\Psi_0$ in (S.14), but the GR requires analyzing $\Psi_1$. This new term $\Psi_1$ is seemingly complicated and its positivity is highly non-trivial, but fortunately, we found the Lemma A.1, that is, $\Psi_1$ is associated with a change of the training loss (i.e., $b$). This evaluation of $\Psi_1$ could potentially be useful on further studies on DLNs. We further emphasize this contribution in the revised manuscript.
>
> >A more general theory, for instance, for locally smooth objectives, could be of more interest.
>
> We agree with the necessity of a general theory, and this is actually what we mentioned in the 3rd paragraph of Section 5.  For example, Ma & Ying 2021 investigated such a locally smooth function on regularizing first-order derivatives of the network output (not the loss). Extending this direction to GR may be interesting. However, note that the performance of GR depends on training algorithms and thus dynamics as we showed. Thus, it seems still challenging to construct a general theory, and our work gives a concrete starting point for further studies on both experiment and theory.
>
> Second, let us reply to your comments on other topics on the algorithmic part:
>
>  >1. intuition behind the efficiency of finite-difference approximation …  not explained why in this case this approximation happens to be better
>
> Our insufficient explanation might cause some confusion. We claimed that the finite difference computation leads to better generalization, but *not that approximation is better*. The estimation of the correct gradient (equivalent to DB) does not lead to better generalization in the case of GR and this is the most interesting point of our work. The following analogy might be helpful as well: consider the usual gradient descent. Taking the small step size leads to the gradient flow which decreases the loss correctly. However, we know that in the practice of deep learning, discrete update with a large step size works better.
>
> > 2. The reduction of double backpropagation to forward passes is not a critical improvement
>
> As we can see in Figure 1, for example, F-GR works more than twice as fast as DB in the training of ResNet-18. Certainly, it does not change the order of the computational complexity, but this improvement seems helpful to train deep networks as large as possible.
>
> >The results do not become much stronger together either, since they are on orthogonal topics
>
> Efficiency and performance might be orthogonal topics, but both are necessary for practice, especially, in the current situation where no one examined either of them.
>  Actually, Reviewer jCXs evaluated this broad perspective as the strength of our work. Although Section 5 is further discussion on different learning methods implicitly connected to the GR,  but as Reviewer hFFP acknowledged, we expect  “it brings some interesting insights that could potentially unify several existing methods”.
>
> Finally, our reply to your question on the coefficient:
>
> >why is $\frac{\gamma}{2}R(\theta)$ used in equation (1) instead of $\frac{\gamma}{4}R(\theta)$?
>
> This is because factor 1/4 is unique to implicit GR [Barrett & Dherin 2021, Smith et al. 2021], which originated from the backward error analysis of the Euler method. Roughly speaking, they map the discrete time-update of the usual gradient descent (w/o GR term) into the continual-time counterpart. In this mapping, we need a Taylor expansion of the gradient and the second order term appears and acts as GR term. As you know, the second order term of the Taylor expansion has a coefficient of 1/2, and the implicit GR term $H\nabla \mathcal{L} \times 1/2 = \nabla R(\theta)/4$.
> In contrast, we focus on explicit GR and there is no necessity to multiply 1/2.

---

> > ### Comment · Reviewer_V6uG · 2022-12-02
> > **Some confusion resolved, other issues remain**
> >
> > I thank the authors for clarifying that the main interest is in the counter-intuitive empirical result that estimating GD accurately doesn't give better generalization. I raise my score as it makes more sense to me now. However, I still find this mostly unexplained and I wish the authors could find a way to justify why this phenomenon is general or at least find reasons behind it on the considered benchmarks.

---

### Public Comment · ~Zhiwei_Jia1 · 2022-11-11
**Related Work**

Hi authors, I like your work on using the finite-difference computation to efficiently regularize the gradient norm term. Please consider citing the following paper that applies gradient regularization similarly by finite-difference computation.

[1] Information-Theoretic Local Minima Characterization and Regularization

ICML 2020

Zhiwei Jia, Hao Su

---

> ### Author Response · Authors · 2022-11-19
> **Thanks for the reference**
>
> In the revised manuscript, we have cited your work as one of the studies that empirically confirmed the generalization improvement. We think that our work gives an explanation as to why the finite difference worked well, and how it will be improved furthermore (i.e., using the forward computation with a larger ascent step).

---

### Decision · Program_Chairs · 2023-01-20

**Decision:**

Reject

**Justification For Why Not Higher Score:**

Please see the above.

**Justification For Why Not Lower Score:**

N/A

**Metareview: Summary, Strengths And Weaknesses:**

Gradient regularization was proposed to improve generalization by finding flatter minima in previous work.  The authors propose a computationally efficient approximation to gradient regularization, using a finite-difference method, evaluate it experimentally, and analyze its effect on the inductive bias in the case of diagonal linear models.

The consensus view was that the paper has some interesting ideas, but that the experiments did not strongly support the claims of the paper, that the theoretical results were difficult to interpret and not clearly explained, and that the theoretical analysis was somewhat incremental.